# SIRT7 is a histone desuccinylase that functionally links to chromatin compaction and genome stability

Lei Li[1,*], Lan Shi[1,*], Shangda Yang[2], Ruorong Yan[1], Di Zhang[1], Jianguo Yang[1], Lin He[1], Wanjin Li[1], Xia Yi[1], Luyang Sun[1], Jing Liang[1], Zhongyi Cheng[3], Lei Shi[2], Yongfeng Shang[1,2] & Wenhua Yu[1]

Although SIRT7 is a member of sirtuin family proteins that are described as $NAD^+$-dependent class III histone deacetylases, the intrinsic enzymatic activity of this sirtuin protein remains to be investigated and the cellular function of SIRT7 remains to be explored. Here we report that SIRT7 is an $NAD^+$-dependent histone desuccinylase. We show that SIRT7 is recruited to DNA double-strand breaks (DSBs) in a PARP1-dependent manner and catalyses desuccinylation of H3K122 therein, thereby promoting chromatin condensation and DSB repair. We demonstrate that depletion of SIRT7 impairs chromatin compaction during DNA-damage response and sensitizes cells to genotoxic stresses. Our study indicates SIRT7 is a histone desuccinylase, providing a molecular basis for the understanding of epigenetic regulation by this sirtuin protein. Our experiments reveal that SIRT7-catalysed H3K122 desuccinylation is critically implemented in DNA-damage response and cell survival, providing a mechanistic insight into the cellular function of SIRT7.

[1] Key Laboratory of Carcinogenesis and Translational Research (Ministry of Education), Department of Biochemistry and Molecular Biology, School of Basic Medical Sciences, Peking University Health Science Center, Beijing 100191, China. [2] 2011 Collaborative Innovation Center of Tianjin for Medical Epigenetics, Tianjin Key Laboratory of Medical Epigenetics, Department of Biochemistry and Molecular Biology, School of Basic Medical Sciences, Tianjin Medical University, Tianjin 300070, China. [3] Jingjie PTM BioLab Co. Ltd., Hangzhou Economic and Technological Development Area, Hangzhou 310018, China. * These authors contributed equally to this work. Correspondence and requests for materials should be addressed to Y.S. (email: yshang@hsc.pku.edu.cn) or to W.Y. (email: yuwenhua@hsc.pku.edu.cn).

Silent information regulator 2 (Sir2) proteins, or sirtuins, were originally discovered for their role in transcriptional repression of several genomic loci in *Saccharomyces cerevisiae*[1]. Mammalian genomes encode seven members of the sirtuin family, SIRT1–7, all possessing a highly conserved catalytic domain and a nicotinamide adenine dinucleotide ($NAD^+$)-binding site while exhibiting different subcellular localization, enzymatic activity, molecular target(s) and tissue specificity[2]. Intriguingly, although SIRT proteins have been described as class III histone deacetylases (HDACs)[3], recent studies suggest that these proteins might possess additional enzymatic activities. For example, it is reported that SIRT3 acts as a decrotonylase to regulate histone lysine crotonylation and gene transcription[4], and that SIRT6 is able to remove fatty acyl modification on lysine (K)19 and K20 of tumour necrosis factor α (ref. 5), while SIRT5, a well-characterized mitochondrial sirtuin protein[6], is shown to negatively regulate several acylations, including succinylation[7], malonylation[8] and glutarylation[9] of both intra- and extra-mitochondrial proteins.

SIRT7 has been identified in the nucleolus and reported to regulate RNA polymerase I transcription[10]. Subsequent studies found that SIRT7 acts as an $NAD^+$-dependent H3K18 deacetylase[11]. In addition, SIRT7 has also been reported to target several non-histone proteins, including p53 (ref. 12), PAF53 (ref. 10), NPM1 (ref. 13), GABP-β1 (ref. 14) and U3-55k (ref. 15) for deacetylation, and has been implicated in hepatic lipid metabolism[16], mitochondrial homeostasis[17] and adipogenesis[18]. However, the enzymatic activity and cellular function of SIRT7 needs further elucidation.

Nucleosome is the fundamental repeating units for packing the large eukaryotic genome into the nucleus while still ensuring appropriate access to it. It consists of 147 bp of DNA wrapped around a protein octamer made up of two copies of each of the four histones H2A, H2B, H3 and H4 (ref. 19). There are two structurally and functionally distinct domains in a histone octamer: the globular domain forming the nucleosomal core around which DNA is wrapped and the unstructured tails to which various post-translational modifications are added[20]. The accessibility of the DNA that is coiled around the histone octamer is a critical parameter for processes such as transcription, replication, recombination and DNA repair. Among various factors that control DNA accessibility, histone modification represents a prominent mechanism by which the nucleosome plasticity is regulated and chromatin configuration is shaped[21,22].

A plethora of histone modifications have been described and, in last decade, various histone modifications, including phosphorylation, acetylation, methylation, ubiquitination, sumoylation and ADP-ribosylation have been the subjects of extensive study in the field of epigenetics[23]. It is believed that factors involved in the deposition (writer), binding (reader) and removal (eraser) of these histone modifications (marks) are at the epicentre of the regulatory circuits controlling the chromatin dynamics[22–24]. It is proposed that various histone modifications in combination constitute distinct histone languages to encode for different chromatin-related events[25]. In effect, chromatin modifiers (writers or erasers) act in an interdependent manner and coordinated fashion to load or remove histone marks to control the chromatin configuration and to determining the biological consequence[22–24]. Accordingly, identification and functional characterization of these chromatin modifiers have been the major theme in the understanding of epigenetic regulation. Strikingly, recent studies identified a series of new types of histone modifications, including biotinylation, citrullination, crotonylation, glutathionylation, propionylation, malonylation and succinylation[9,26–29], adding to the complexity of the already sophisticated epigenetic regulatory network.

Ultimately, the understanding of the biological significance of these new modifications is still dependent on identification of the writers, readers and erasers of these histone marks.

In this study, we report that SIRT7 is an $NAD^+$-dependent histone desuccinylase. We show that SIRT7 is recruited to double-strand break (DSB) sites in a PARP1-dependent manner and catalyses desuccinylation of H3K122 at DSB sites, thereby promoting chromatin condensation and efficient DSB repair. We demonstrate that depletion of SIRT7 impaired chromatin compaction and DNA repair, and sensitized cells to genotoxic stresses.

## Results

**SIRT7 is a histone desuccinylase**. As stated above, sirtuin family proteins were initially characterized as class III histone deacetylases[3] but recently reported to possess additional deacylation activities[7,8,30]. To better understand the enzymatic activity and cellular function of SIRT7, we started by profiling the expression of SIRT7 in various cell lines. The results showed that SIRT7 was widely expressed in different cell lines (Fig. 1a). SIRT7 was then stably knocked down in SIRT7 highly expressed MCF-7 cells, and the alteration in the levels of histone lysine crotonylation, succinylation and malonylation was assessed by an integrated approach of stable isotope labelling by amino acids in cell culture (SILAC) and mass spectrometry-based quantitative proteomics (Fig. 1b). Triplicate experiments showed that knockdown (KD) of SIRT7 did not result in evident changes in the level of histone lysine malonylation, and that the level of histone crotonylation rather decreased on SIRT7 depletion (Fig. 1c), suggesting that these histone marks might not be targeted directly by SIRT7. However, highly reproducible increases in the levels of succinylation of H2BK46, H2BK108, H4K31 and H4K77, especially H3K122, were detected on SIRT7 depletion (Fig. 1c), suggesting that SIRT7 is functionally associated with the regulation of histone succinylation.

To strengthen the functional connection between SIRT7 and histone succinylation, we generated three SIRT7 mutants: SIRT7H187Y, substitution of the highly conserved histidine residue (His187) in the predicted catalytic domain with tyrosine[11]; SIRT7S111A, substitution of the highly conserved serine residue (Ser111) in the $NAD^+$-binding pocket with alanine[31,32]; and a double mutant SIRT7H187Y/S111A (SIRT7DM). Wild-type SIRT7 and its mutants were then expressed in HEK293T cells. Histone succinylation was analysed by western blot with antibodies against pan-lysine succinylation. The specificity of these antibodies were verified by dot blot (Supplementary Fig. 1). Overexpression of either wild-type SIRT7 or SIRT7S111A was associated with a decrease in histone H3 lysine pan-succinylation, whereas overexpression of SIRT7H187Y, SIRT7DM or SIRT6 did not result in evident changes in histone H3 pan-lysine succinylation (Fig. 1d and Supplementary Fig. 2a,b). Similarly, overexpression of either wild-type SIRT7 or SIRT7S111A was associated with a decrease in H3K18ac, whereas overexpression of SIRT7H187Y, SIRT7DM or SIRT6 did not result in detectable changes in this modification (Fig. 1d and Supplementary Fig. 2a,b). In addition, KD of SIRT7 led to a more than twofold increase in the average fluorescent intensity of lysine succinylation (Fig. 1e) while overexpression of green fluorescent protein (GFP)-SIRT7 resulted in a decrease in pan-lysine succinylation (Supplementary Fig. 2c). Together, these experiments indicate that SIRT7 regulates histone succinylation in a catalytic activity-dependent manner, supporting a notion that SIRT7 functions as a potential histone desuccinylase.

Since succinylation is a newly identified histone modification and there are no commercially available antibodies against this

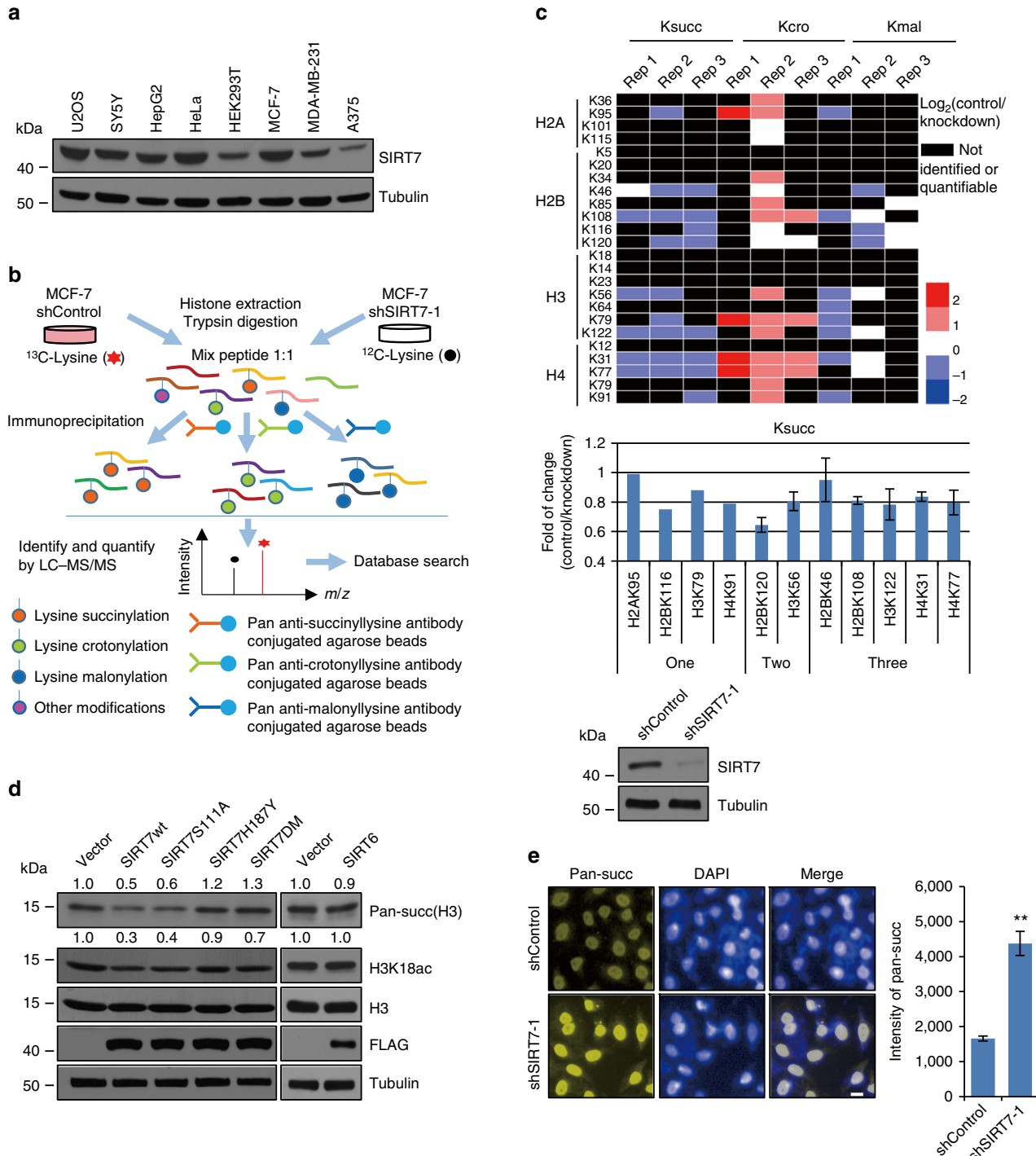

**Figure 1 | SIRT7 regulates histone succinylation.** (**a**) Western blotting analysis of SIRT7 expression in different cell lines. (**b**) The workflow of the integrated SILAC labelling, affinity enrichment and mass spectrometry-based quantitative proteomics to quantify dynamic changes of histone lysine crotonylation (Kcro), succinylation (Ksucc) and malonylation (Kmal) in control or SIRT7-depleted MCF-7 cells. (**c**) Heatmap of the changes in the levels of Kcro, Ksucc and Kmal of histones detected by SILAC labelling and mass spectrometry-based quantitative proteomics. All experiments were carried out in triplicate and the results are presented as base 2 logarithmic value of the ratio of control/KD. Results of detectable succinylation in one, two or three replicates are also presented as histogram. The efficiency of SIRT7 KD in MCF-7 cells was monitored by western blotting. (**d**) HEK293T cells were transfected with wild-type SIRT7, SIRT7 mutants or SIRT6. Histones were extracted and pan-succinylation of H3 and H3K18ac were analysed by western blotting. The bands were quantified with ImageJ software. The numbers indicate the relative levels of the indicated modifications. Whole-cell lysate was prepared for monitoring the efficiency of overexpression of SIRT7 and SIRT6 by western blotting. (**e**) High-throughput microscopic analysis of the mean relative fluorescence intensity of pan-succinylation in control or SIRT7-depleted MCF-7 cells. Scale bar, 10 μm. Each bar represents the mean ± s.d. for triplicate experiments. **$P < 0.01$ (two-tailed unpaired Student's $t$-test).

modification yet, we thus generated polyclonal antibodies against two succinylation modifications, H3K122 succinylation (H3K122succ) and H2BK120 succinylation (H2BK120succ), based on our observation that SIRT7 KD resulted in the most evident changes in the levels of H3K122succ and H2BK120succ in the SILAC/liquid chromatography coupled tandem mass spectrometry (LC–MS/MS) experiments. The specificity of these antibodies were verified by peptide competition assays using synthesized peptides (Fig. 2a and Supplementary Fig. 3a–c). In addition, FLAG-tagged wild-type H3 (FLAG-H3wt) or lysine 122-mutated and succinylation-resistant[33] H3 mutant

(FLAG-H3K122R) were also used to validate the specificity of the antibodies against H3K122succ (Fig. 2a). Overexpression of SIRT7 led to a decrease in H3K122succ level in HEK293T cells and KD/knockout of SIRT7 resulted in an increase in H3K122succ level in MCF-7, HCT116, HeLa or U2OS cells, whereas the levels of H2BK120succ and H3K122ac showed no obvious changes in these cells, regardless of overexpression or KD or knockout of SIRT7 (Fig. 2b and Supplementary Fig. 4a–c). Meanwhile, overexpression of SIRT6 in HEK293T cells or KD of SIRT6 in MCF-7 cells did not result in evident changes in H3K122succ and H2BK120succ (Fig. 2b and Supplementary

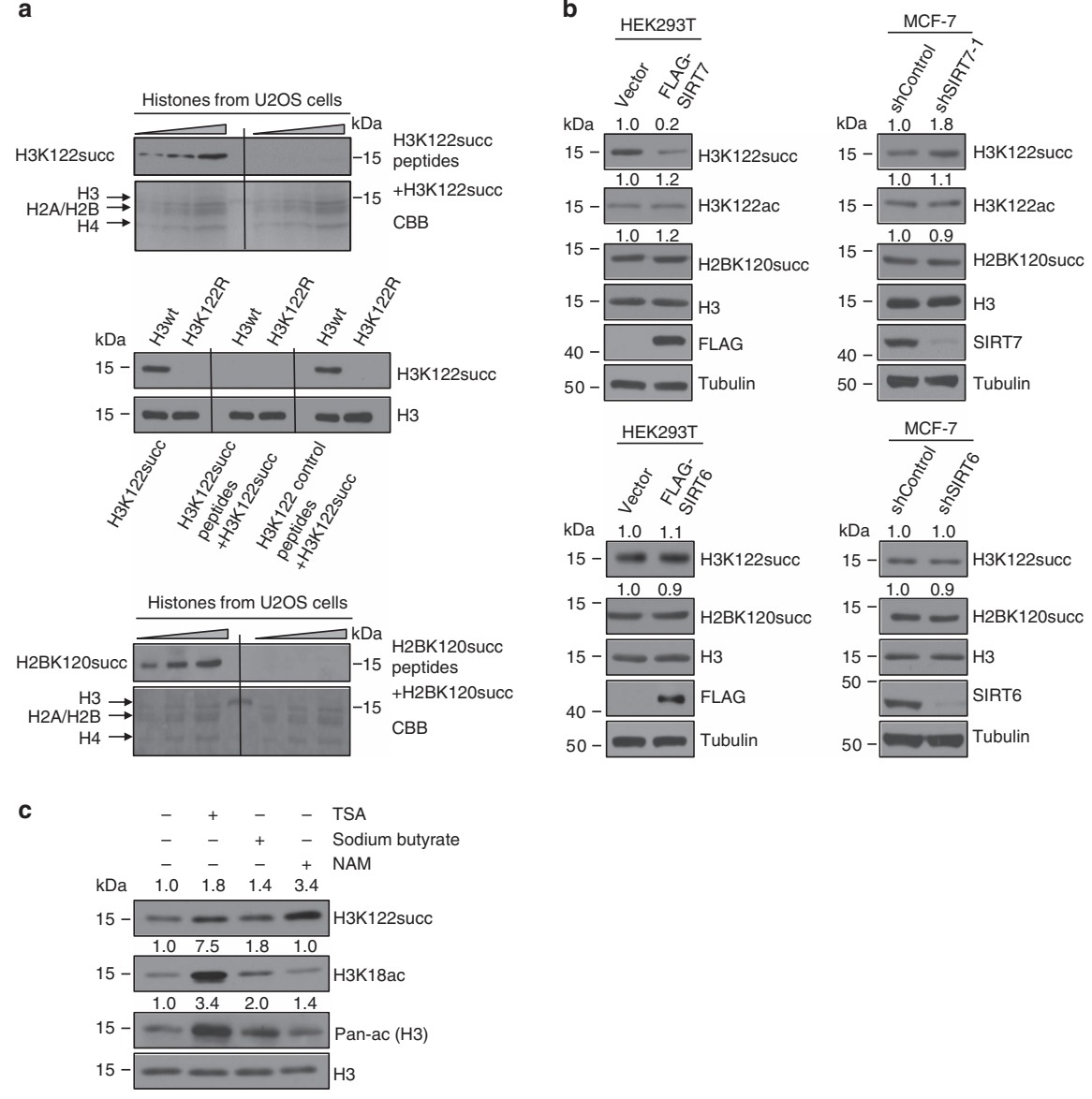

**Figure 2 | SIRT7 regulates histone H3K122 succinylation *in vivo*. (a)** Confirmation of the specificity of H3K122succ (upper) or H2BK120succ (lower) antibodies by peptide competition experiments. Different amounts of soluble histones extracted from U2OS cells were resolved on SDS–polyacrylamide gel electrophoresis (SDS–PAGE) gels, probed with anti-H3K122succ or anti-H2BK120succ with or without excessive H3K122succ or H2BK120succ peptides. The extracted histones were also resolved on SDS–PAGE and stained with Coomassie brilliant blue (CBB). FLAG-H3wt or FLAG-H3K122R was purified from U2OS cells, resolved on SDS–PAGE gels and probed with the anti-H3K122succ antibody alone or anti-H3K122succ antibody pre-adsorbed with H3K122succ peptides or H3K122 control peptides, with H3 analysis as an internal control (middle). (**b**) SIRT7 or SIRT6 was overexpressed in HEK293T cells or knocked down in MCF-7 cells. Histones were extracted and the succinylation and acetylation were analysed by western blotting with the indicated antibodies. The efficiency of overexpression or KD of SIRT7 and SIRT6 was monitored by western blotting of whole-cell lysate, with corresponding antibodies. (**c**) MCF-7 cells were treated with 10 μM trichostatin A (TSA), 4 mM of sodium butyrate or 10 mM NAM. Twenty four hours later, cells were collected and soluble histones were prepared and subjected to western blotting with antibodies as indicated.

Fig. 4a). As stated above, the sirtuin proteins utilize $NAD^+$ as a cofactor for their catalytic activity. To further substantiate the role of SIRT7 in histone desuccinylation, MCF-7 cells were treated with the general sirtuin inhibitor nicotinamide (NAM), or the HDAC inhibitors trichostatin A or sodium butyrate. Although H3K122succ levels increased slightly following trichostatin A and sodium butyrate treatment, NAM treatment was associated with a marked increase in H3K122succ level (Fig. 2c and Supplementary Fig. 4d). Collectively, these results suggested that SIRT7 is a $NAD^+$-dependent H3K122succ desuccinylase.

To further support this proposition, FLAG-SIRT7wt, FLAG-SIRT7H187Y, FLAG-SIRT5 or FLAG-SIRT6 were expressed in and purified from HEK293T cells (Fig. 3a) for in vitro desuccinylation assays. Incubation of H3K122succ peptides with these proteins revealed that the level of H3K122succ significantly decreased when FLAG-SIRT7wt and $NAD^+$ was included (Fig. 3b). The level of H3K122succ did not change when FLAG-SIRT7H187Y was used (Fig. 3b). Notably, SIRT5 and SIRT6 also showed more or less H3K122succ desuccinylase activity (Fig. 3b). In vitro desuccinylation assays were also performed using calf thymus histones as substrates. Incubation of calf thymus histones with FLAG-SIRT7wt resulted in an overt decrease in the level of H3K122succ in a dose-dependent manner, an effect could be abolished by NAM, whereas no obvious changes were observed in the levels of H2BK120succ and H3K122ac (Fig. 3c and Supplementary Fig. 5a). Incubation of calf thymus histones with FLAG-SIRT7H187Y, FLAG-SIRT5 or FLAG-SIRT6 did not result in evident changes in the levels of H3K122succ, H3K122ac and H2BK120succ (Fig. 3c and Supplementary Fig. 5a). The discrepancy concerning H3K122succ desuccinylation by SIRT5 and SIRT6 on H3K122succ peptides versus calf thymus histones is unknown but could be possibly due to the overall structural difference between the synthesized peptides and the natural calf thymus histones. Mass spectrometric analysis also showed that SIRT7 exhibited robust H3K122succ desuccinylase activity while had no effect on H2BK120succ, H3K122ac and H3K18ac (Fig. 3d–f). In vitro desuccinylation assays were also performed with mononucleosomes isolated from HeLa cells, consistent with the results obtained with calf thymus histones, incubation of mononucleosomes with FLAG-SIRT7wt resulted in a marked and $NAD^+$-dependent decrease in the level of H3K122succ, an effect that could be abolished by NAM, whereas FLAG-SIRT7H187Y had no evident effect on this modification (Fig. 3g and Supplementary Fig. 5b). In addition, our data indicate that SIRT7 showed no substrate preference between nucleosomes and calf thymus histones (Supplementary Fig. 5c). Together, these results further demonstrated that SIRT7 is a $NAD^+$-dependent H3K122succ desuccinylase.

**SIRT7 is transiently recruited to DNA-damage sites**. To explore the biological significance of SIRT7-mediated histone desuccinylation, we next used immunopurification and mass spectrometry to identify proteins that are potentially associated with SIRT7 in vivo. The results showed that SIRT7 could be co-purified with a number of proteins, including DNA-PKcs, RAD50, PARP1, Ku80 and Ku70, that are known to be involved in DNA repair[34], as well as proteins implicated in other cellular processes such as chromatin remodelling and ribosomal biogenesis (Fig. 4a).

The co-purification of DNA repair proteins with SIRT7 suggests that SIRT7 may play a role in DNA-damage response. In this regard, it is interesting to note that it has been reported that Sirt7-deficient primary cardiomyocytes exhibit an increase in basal apoptosis and a significantly diminished resistance to

oxidative and genotoxic stress[12], and that SIRT7 was shown to promote cellular survival following genomic stress[13]. In addition, it was noted that K122Q and K122A mutants of H3K122 were much more sensitive to DNA-damaging reagents than wild type[35]. To test the hypothesis that SIRT7 is involved in DNA repair and in the maintenance of genome integrity, ultraviolet laser microirradiation system was utilized to generate localized DNA damage[36] in GFP-SIRT7-expressing human U2OS cells. Real-time imaging of living cells showed accumulation of GFP-SIRT7 at the sites of DNA damage immediately after microirradiation. The fluorescence intensified quickly, reaching to maximum at about 4–5 min, and attenuated thereafter, receding to pre-damage level at about 15 min (Fig. 4b), suggesting that the recruitment of SIRT7 to DNA-damage sites is a transient process. Notably, the accumulation of SIRT7 at DSBs represented only a fraction of the nuclear pool of this protein, the bulk of which remained concentrated in the nucleolus (Fig. 4b). Time-lapse analysis revealed that the accumulation of GFP-SIRT7 at the DSB sites reached half-maximum within 1 min after microirradiation (Fig. 4b). Collectively, these results indicate that SIRT7 is recruited to DNA-damage sites rapidly and transiently, suggesting that SIRT7 might function at an early and priming stage of DNA-damage response.

To gain a mechanistic insight into the rapid and transient recruitment of SIRT7 to DNA damage, we next investigated the functional relationship between SIRT7 and the two important factors involved in the early steps of DNA-damage response: PARP1 and ATM[37], especially PARP1 was identified to be co-purified with SIRT7 (Fig. 4a). To this end, U2OS cells were pre-treated with PARP1 inhibitor PJ-34 (ref. 38), or ATM inhibitor KU-55933 (ref. 39), followed by laser microirradiation and immunostaining. Immunofluorescent microscopy showed that, consistent with GFP-SIRT7, endogenous SIRT7 in U2OS cells without pre-treatment was efficiently recruited to the sites of laser-induced DNA breaks 5 min after laser microirradiation, as marked by γH2AX staining, whereas chemical inhibition of PARP1 enzyme, but not ATM, resulted in a complete abrogation of SIRT7 accumulation at laser-induced damage sites (Fig. 4c). Consistently, KD of PARP1 also resulted in a complete abrogation of SIRT7 accumulation at laser-induced damage sites (Supplementary Fig. 6a,b). Co-immunoprecipitation experiments showed that PARP1 was efficiently co-immuno-precipitated with SIRT7, and, remarkably, the interaction was intensified when cells were exposed to ionizing radiation (IR) (Fig. 4d). These results suggest that the recruitment of SIRT7 to DSB sites is dependent on PARP1.

**SIRT7-catalyzed H3K122 desuccinylation in DSB repair**. In order to explore the functional significance of the recruitment of SIRT7 to DNA-damage sites, we examined the effect of SIRT7 on the repair efficiency of two major DSB repair pathways, non-homologous end joining (NHEJ) and homologous recombination (HR), using EJ5-GFP-HEK293 and DR-GFP-U2OS systems, respectively, as we described previously[38]. For NHEJ repair, depletion of SIRT7 expression was associated with a dramatic reduction of the relative percentage of GFP-positive cells by about 90%, which was comparable to the effect of depletion of Ku80, an essential component of NHEJ repair[34] (Fig. 5a and Supplementary Fig. 7a). The KD efficiency of SIRT7 and Ku80 were monitored by western blotting (Fig. 5a). Meanwhile, the effect of SIRT7 on HR repair was evaluated in U2OS cells using the DR-GFP system. The results showed that depletion of SIRT7 resulted in a marked decrease in the percentage of GFP-positive cells, an effect comparable to that of KD of BRCA1, a key regulator of HR repair pathway[34] (Fig. 5b and Supplementary

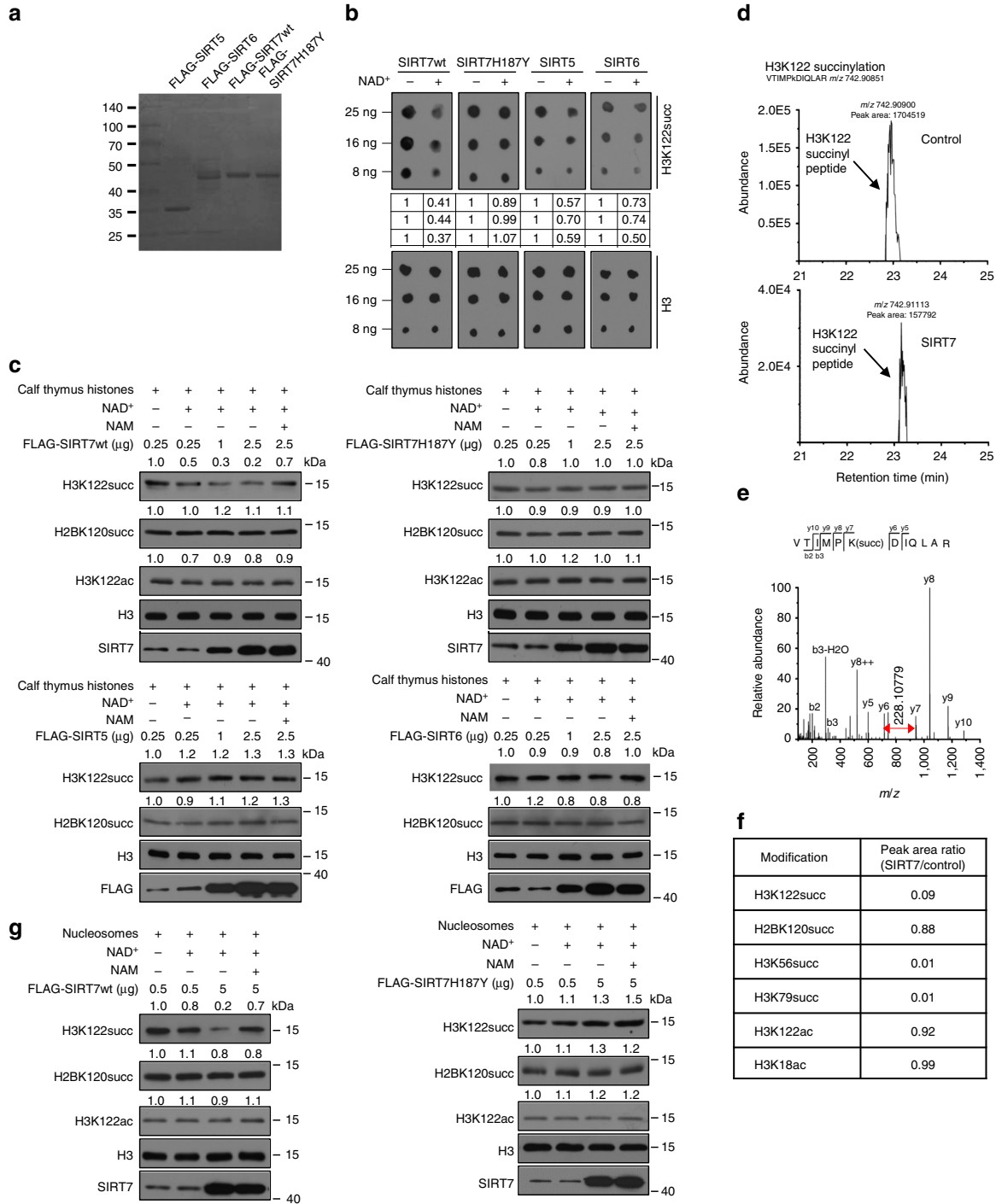

**Figure 3 | SIRT7 catalyses histone H3K122 desuccinylation *in vitro*.** (**a**) FLAG-SIRT5, FLAG-SIRT6, FLAG-SIRT7wt or FLAG-SIRT7H187Y was expressed in and purified with anti-FLAG M2 affinity gel from HEK293T cells and stained with Coomassie brilliant blue. (**b**) *In vitro* desuccinylation assays with synthesized H3K122succ peptides. Two micrograms of purified FLAG-SIRT7wt, FLAG-SIRT7H187Y, FLAG-SIRT5 or FLAG-SIRT6 were incubated with 500 ng H3K122succ peptides in the presence or absence of 1.0 mM NAD$^+$. The reaction mixtures containing 8, 16 or 25 ng peptides were boiled and subjected to dot blot analysis with anti-H3K122succ or anti-H3. The dots were quantified by densitometry with ImageJ software. The numbers indicate the relative levels of the indicated modifications. (**c**) *In vitro* desuccinylation assays with calf thymus histones. Different amounts of purified FLAG-SIRT7wt, FLAG-SIRT7H187Y, FLAG-SIRT5wt or FLAG-SIRT6wt were incubated with 1 µg calf thymus histones in the presence of 1.0 mM NAD$^+$ and/or 10 mM NAM. The reaction mixtures were boiled and analysed by western blotting with the indicated antibodies. (**d**) The base peaks of H3K122succ in control and SIRT7-treated calf thymus histones. The peak areas were used for the quantification of H3K122succ in the two samples. (**e**) The MS/MS spectra for the identification of H3K122succ. b and y ions indicate peptide backbone fragment ions containing the N and C terminal, respectively. + + indicates doubly charged ions. (**f**) The quantification ratios of several succinylation and acetylation sites in histones by comparing the peak areas in SIRT7-treated and control samples. (**g**) *In vitro* desuccinylation assays with mononucleosomes. Different amounts of purified FLAG-SIRT7wt or FLAG-SIRT7H187Y were incubated with 1 µg HeLa cell-derived mononucleosomes in the presence or absence of 1.0 mM NAD$^+$ and/or 10 mM NAM. The reaction mixture was analysed by western blotting with the indicated antibodies.

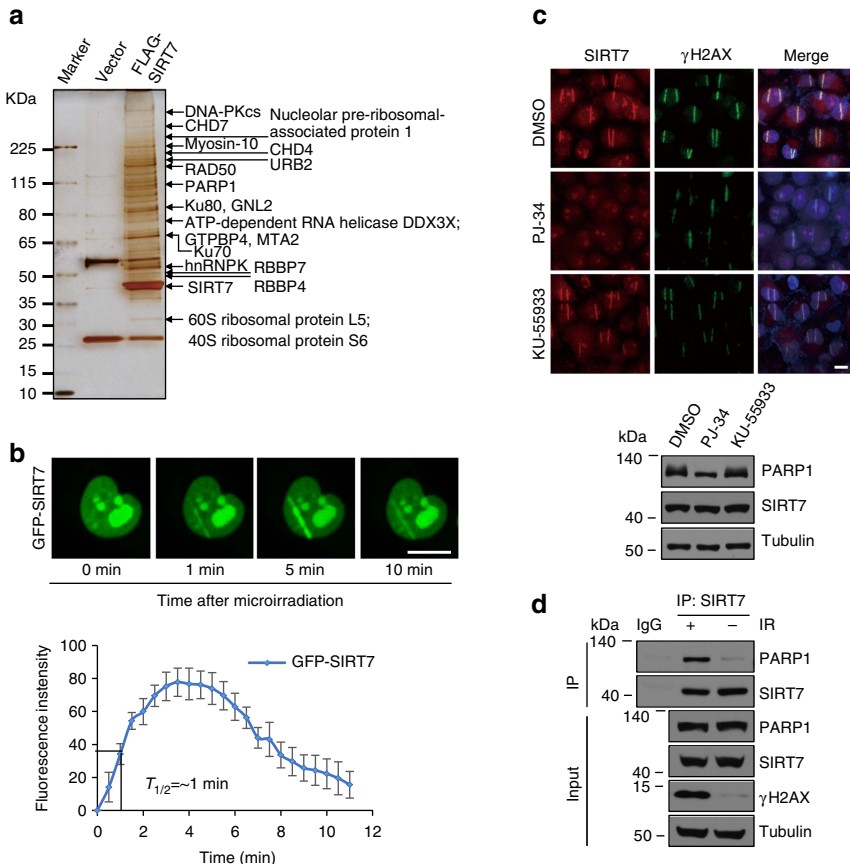

**Figure 4 | SIRT7 is transiently recruited to DNA-damage sites in a PARP1-dependent manner. (a)** Whole-cell extracts from HEK293T cells expressing vector or FLAG-SIRT7wt were subjected to affinity purification with anti-FLAG M2 affinity gel. The purified protein complex was resolved on SDS–polyacrylamide gel electrophoresis and silver stained. The bands were retrieved and analysed by mass spectrometry. **(b)** U2OS cells stably expressing GFP-SIRT7 were subjected to laser microirradiation using micropoint system and analysed for the accumulation of GFP-SIRT7 in DSBs by fluorescent microscopy (upper). Scale bar, 10 μm. The real-time recruitment of GFP-SIRT7 was also analysed in 30 independent cells (lower). Error bars indicate mean ± s.e.m. **(c)** U2OS cells were pre-treated with PARP1 inhibitor PJ-34 or ATM inhibitor KU-55933 for 1 h and subjected to laser microirradiation and immunofluorescent analysis of SIRT7 and γH2AX at 5 min after microirradiation (upper). Scale bar, 10 μm. The expression of SIRT7 was analysed by western blotting (lower). **(d)** U2OS cells were exposed to 6 Gy of IR. Whole-cell lysate was immunoprecipitated with antibodies against SIRT7 followed by immunoblotting with the indicated antibodies.

Fig. 7b). Together, these results indicate that SIRT7 is required for efficient repair of DSBs.

To further investigate whether the catalytic activity of SIRT7 is required for efficient DSB repair, rescue experiments were performed by using SIRT7 siRNA-1 resistant SIRT7 (rSIRT7). The results revealed that rSIRT7wt was able to rescue the decreased NHEJ repair efficiency induced by depletion of endogenous SIRT7, whereas rSIRT7H187Y was not (Fig. 5c). Similarly, rSIRT7wt expression was associated with a restoration of HR repair in SIRT7-depleted cells, whereas rSIRT7H187Y was not (Fig. 5c). Furthermore, quantitative chromatin immunoprecipitation (qChIP) assays in DR-GFP-U2OS cells showed that both rSIRT7wt and rSIRT7H187Y were effectively recruited to break sites, with rSIRT7H187Y even exhibiting a stronger binding capacity than rSIRT7wt (Fig. 5d), excluding the possibility that failing to rescue the decreased NHEJ and HR repair efficiency by rSIRT7H187Y resulted from deficient recruitment of rSIRT7H187Y to DSB sites. The efficiency of KD of SIRT7 and overexpression of I-SceI, rSIRT7wt, or rSIRT7H187Y was monitored by western blotting (Fig. 5e). The specificity of SIRT7 KD by siRNA was also validated (Fig. 5f). Collectively, these results suggest that the catalytic activity of SIRT7 is required for its function in DNA repair.

To investigate whether SIRT7-catalyzed H3K122succ desuccinylation was involved in DNA-damage response, MCF-7 cells were treated with DNA damaging reagents etoposide (VP16) or camptothecin (CPT), histone succinylation and acetylation levels were measured by western blotting. The results showed that treatment with either VP16 or CPT resulted in evident decrease in the levels of H3K122succ and H3 pan-succinylation, whereas the levels of H3K122ac, H2BK120succ, H3K18ac and H3 pan-acetylation showed no obvious changes (Fig. 6a). However, when SIRT7 was knocked down in MCF-7 cells, treatment with CPT was no longer associated with decreases in H3K122succ level (Fig. 6b and Supplementary Fig. 8a). Similar result was also observed in U2OS cells (Fig. 6b and Supplementary Fig. 8b). Furthermore, U2OS cells were treated with IR and histones were extracted at different time points after IR for western blotting analysis of H2K122succ. We found that, compared with that at 0 h, the level of H3K122succ was significantly decreased at 1 h after IR, and gradually recovered at 4 h after IR, meanwhile no obvious changes were detected in the levels of H3K122ac and H2BK120succ (Fig. 6c and Supplementary Fig. 8c). However, when SIRT7 was depleted, treatment with IR no longer resulted in decrease in H3K122succ level, the levels of H3K122ac and H2BK120succ also had no obvious changes (Fig. 6c and Supplementary Fig. 8c). These results suggest that

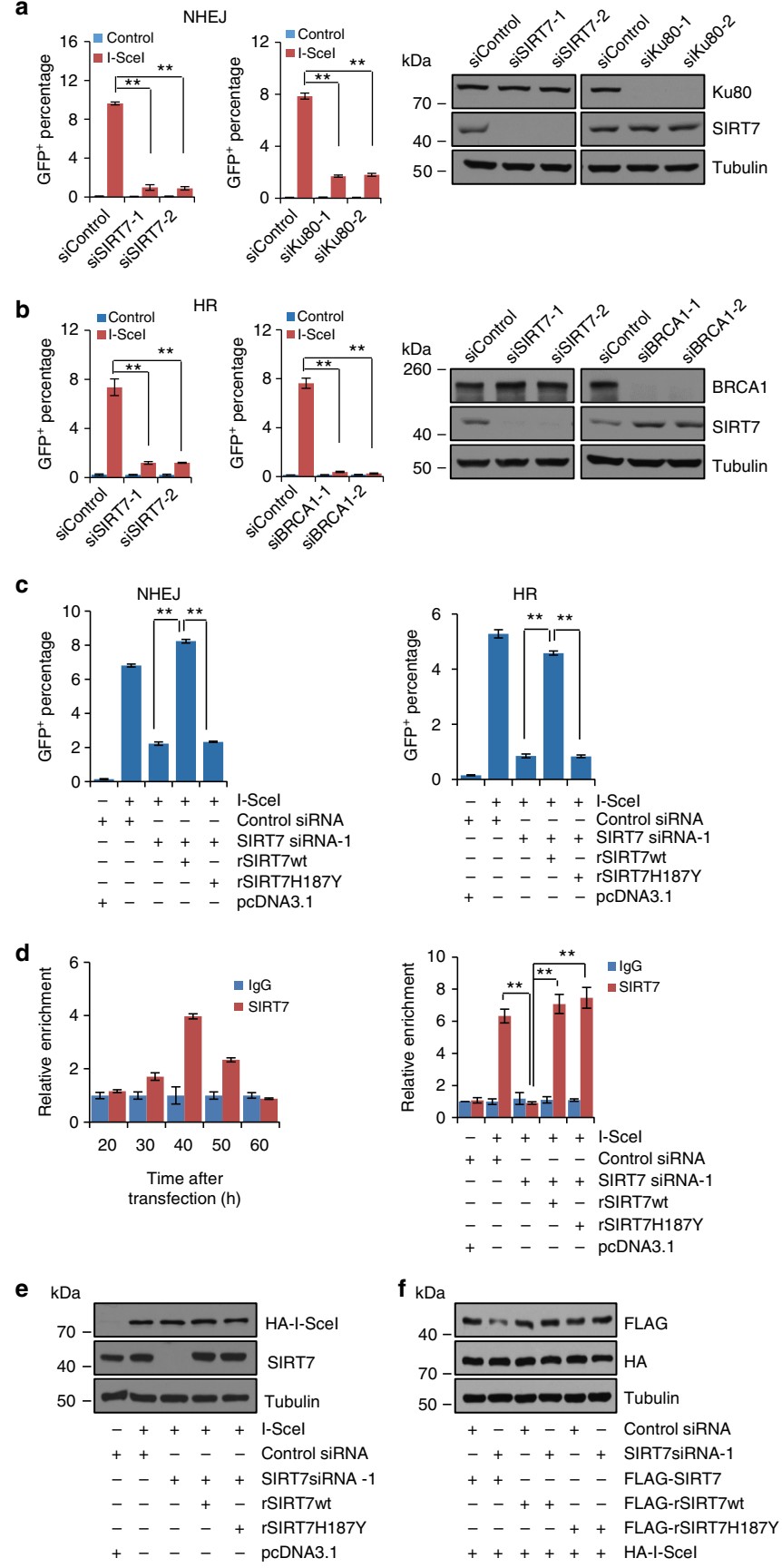

DNA-damage reagent-induced decrease of H3K122succ was functionally linked to SIRT7.

We then performed laser microirradiation and immunofluorescent assays using anti-H3K122succ to address whether the decrease of H3K122succ is SIRT7- and PARP1-dependent and specific to the DSB sites. In control U2OS cells, H3K122succ level at DSB sites decreased 5 min after microirradiation, while in SIRT7-depleted U2OS cells, H3K122succ level at DSB sites did not change (Fig. 6d,f). We then analysed H3K122succ level at DSB sites in U2OS cells transfected with siControl or siPARP1. The results showed that in siControl cells H3K122succ level at DSB sites decreased 5 min after microirradiation, while in siPARP1 cells H3K122succ level at DSB sites had no detectable change (Fig. 6e,f). It is worthy of noting that ultraviolet laser microirradiation and ionizing radiation are different DNA insults. Thus, these stimuli-evoked DNA-damage responses might be different, at least in terms of time course and factor recruitment, which could potentially contribute to the difference of the dynamic changes of H3K122succ level observed in these microirradiation experiments versus that in IR experiments in Fig. 6c. These results suggest that SIRT7-mediated H3K122succ desuccinylation after microirradiation is PARP1-dependent and specific to DSB sites.

qChIP assays were then performed and the level of H3K122succ at DNA-damage sites was measured using HR repair system in DR-GFP-U2OS cells. The results showed that transfection of the cells with I-SceI led to a marked decrease in the level of H3K122succ around the break sites (Fig. 6g). However, when SIRT7 was knocked down, I-SceI transfection no longer resulted in a decrease in the level of H3K122succ at the break sites (Fig. 6g), and ectopic expression of rSIRT7wt, but not rSIRT7H187Y, could fully restore the downregulated level of H3K122succ (Fig. 6g). These results further indicate that SIRT7 is recruited to DSB sites and catalysed H3K122succ desuccinylation therein.

**H3K122 desuccinylation by SIRT7 in chromatin condensation.** The lateral surface site H3K122 has been implicated in DNA-damage repair[35], and H3K122ac has been reported to contribute to a relaxed chromatin state associated with active transcription[40]. In comparison, succinylation is a more acidic modification, which could change the charge on lysine from $+1$ to $-1$ under physiological conditions[30]. Therefore, we speculated that succinylation of H3K122 is associated with chromatin relaxation while desuccinylation of H3K122succ linked to a more condensed chromatin state. To test this hypothesis, we generated FLAG-tagged constructs of wild-type H3 (H3wt) or H3 mutants H3K122E, which mimic the negative charge state of the succinyl group[33], and non-succinylated H3K122R (ref. 33). The chromatin

structure of U2OS cells stably transfected with these constructs was analysed by micrococcal nuclease (MNase) sensitivity assays[41]. We found that compared with stable expression of H3wt, stable expression of H3K122E resulted in an increase, albeit moderately, in MNase accessibility in chromatin, whereas stable expression of H3K122R led to a decrease in MNase sensitivity (Fig. 7a and Supplementary Fig. 9a,b), suggesting that H3K122 succinylation is associated with an increased chromatin accessibility, whereas desuccinylation of H3K122succ is linked to a condensed chromatin state.

Histone–DNA interactions are very stable; histones are only released from chromatin by NaCl concentrations in excess of 1.5 M (refs 42,43). To substantiate the relationship between H3K122succ and chromatin compaction, nucleosome stability assay[44] was performed to investigate the difference of histone–DNA interactions in H3wt, H3K122E and H3K122R stably expressing U2OS cells. Compared with H3wt, the NaCl solubility of histone H3 in H3K122E stably expressing cells increased markedly, while the NaCl solubility of histone H3 in H3K122R cells decreased significantly (Fig. 7b). It suggests that succinylation of H3K122 was associated with a reduction of histone–DNA interactions while desuccinylation of H3K122succ stabilizes the nucleosomes, consistent with our hypothesis that H3K122 succinylation is associated with an increased chromatin accessibility, whereas desuccinylation of H3K122succ is linked to a condensed chromatin state. In addition, cells overexpressing GFP-SIRT7 had smaller nuclear size compared with cells that expressed GFP only (Supplementary Fig. 9c). In agreement with this observation, detailed quantification of proliferation patterns in shControl and shSIRT7 cells demonstrated that although SIRT7 KD group had a similar number of cells in S phase to that in control group, the number of cells in late S phase on SIRT7 KD decreased (Supplementary Fig. 9d). These observations altogether point to a role of SIRT7-meidated H3K122succ desuccinylation in chromatin compaction.

To strengthen this argument, we performed MNase sensitivity assays in control or SIRT7-depleted U2OS cells under treatment of IR. Knockdown of SIRT7 resulted in an, albeit moderate, increase in MNase sensitivity of chromatin, and the effect was significantly augmented following IR treatment (Fig. 7c and Supplementary Fig. 9e,f). These data suggest that loss of function of SIRT7 is associated with chromatin decompaction thus compromises subsequent chromatin relaxation, consistent with a previous report that decompaction of chromatin by overexpression or tethering H3K4 methyltransferase ASH2 dampens DNA-damage response and alters local chromatin dynamics[45]. We also performed nucleosome stability assays in control or SIRT7-depleted U2OS cells under IR treatment. The results showed that DNA damage increased the NaCl

**Figure 5 | The catalytic activity of SIRT7 is required for efficient DSB repair.** (**a**) NHEJ efficiency was determined by FACS in SIRT7- or Ku80-deficient EJ5-HEK293 cells. Each bar represents the mean ± s.d. for triplicate experiments. KD efficiency of SIRT7 and Ku80 was examined by western blotting. (**b**) HR efficiency was determined by FACS in SIRT7- or BRCA1-deficient DR-GFP-U2OS cells. Each bar represents the mean ± s.d. for triplicate experiments. KD efficiency of SIRT7 and BRCA1 was examined by western blotting. (**c**) Rescue experiments for NHEJ or HR deficiency induced by SIRT7 depletion. EJ5-GFP-HEK293 cells (left) or DR-GFP-U2OS cells (right) stably expressing SIRT7 siRNA-1-resistant SIRT7wt (rSIRT7wt) or siRNA-1-resistant SIRT7H187Y (rSIRT7H187Y) were transfected with control siRNA or siSIRT7-1 as indicated. Twenty four hours later, the cells were transfected with pcDNA3.1 vector or I-SceI for 48 h, and collected and analysed by FACS. Each bar represents the mean ± s.d. for triplicate experiments. (**d**) SIRT7 occupancy at chromatin flanking DSB generated by endonuclease I-SceI. DR-GFP-U2OS cells transfected with I-SceI were collected at different time points and subjected to ChIP assay, with antibodies against SIRT7. The final DNA extractions were amplified by quantitative real-time PCR using primer that covers the DNA sequences flanking the I-SceI site. Each bar represents the mean ± s.d. for triplicate experiments (left). Control or DR-GFP-U2OS cells stably expressing rSIRT7wt or rSIRT7H187Y were transfected with control siRNA or siSIRT7-1 as indicated. Twenty four hours later, the cells were transfected with pcDNA3.1 vector or I-SceI for 40 h and subjected to qChIP analysis with antibodies against SIRT7. Each bar represents the mean ± s.d. for triplicate experiments (right). (**e**) The efficiency of SIRT7 KD and overexpression of I-SceI, rSIRT7wt or rSIRT7H187Y in DR-GFP-U2OS cells. Tubulin was analysed as an internal control. (**f**) The KD specificity of SIRT7 siRNA-1 for FLAG-tagged wild-type SIRT7, siRNA-1-resistant rSIRT7wt or rSIRT7H187Y in DR-GFP-U2OS cells. The haemagglutinin (HA)-tagged I-SceI and tubulin were used as loading controls. **P < 0.01 (two-tailed unpaired Student's t-test).

solubility of histones γH2AX, H2AX and H3 within damaged chromatin (Fig. 7d). Importantly, when SIRT7 was depleted, the DNA damage-dependent increase in NaCl solubility of histones was enhanced (Fig. 7d), suggesting that SIRT7-mediated H3K122succ desuccinylation promotes the stability of histone–DNA interactions within nucleosomes during DNA-damage response, thereby creating a condensed chromatin domain.

To further investigate the importance of H3K122succ desuccinylation in chromatin condensation thus efficient DSB repair, analysis of NHEJ repair efficiency in EJ5-GFP-HEK293 cells stable expressing FLAG-H3wt, FLAG-H3K122E or FLAG-H3K122R by fluorescence-activated cell sorting (FACS) showed that expression of H3K122E and H3K122R reduce NHEJ repair efficiency (Fig. 7e and Supplementary Fig. 10a,b). Significant

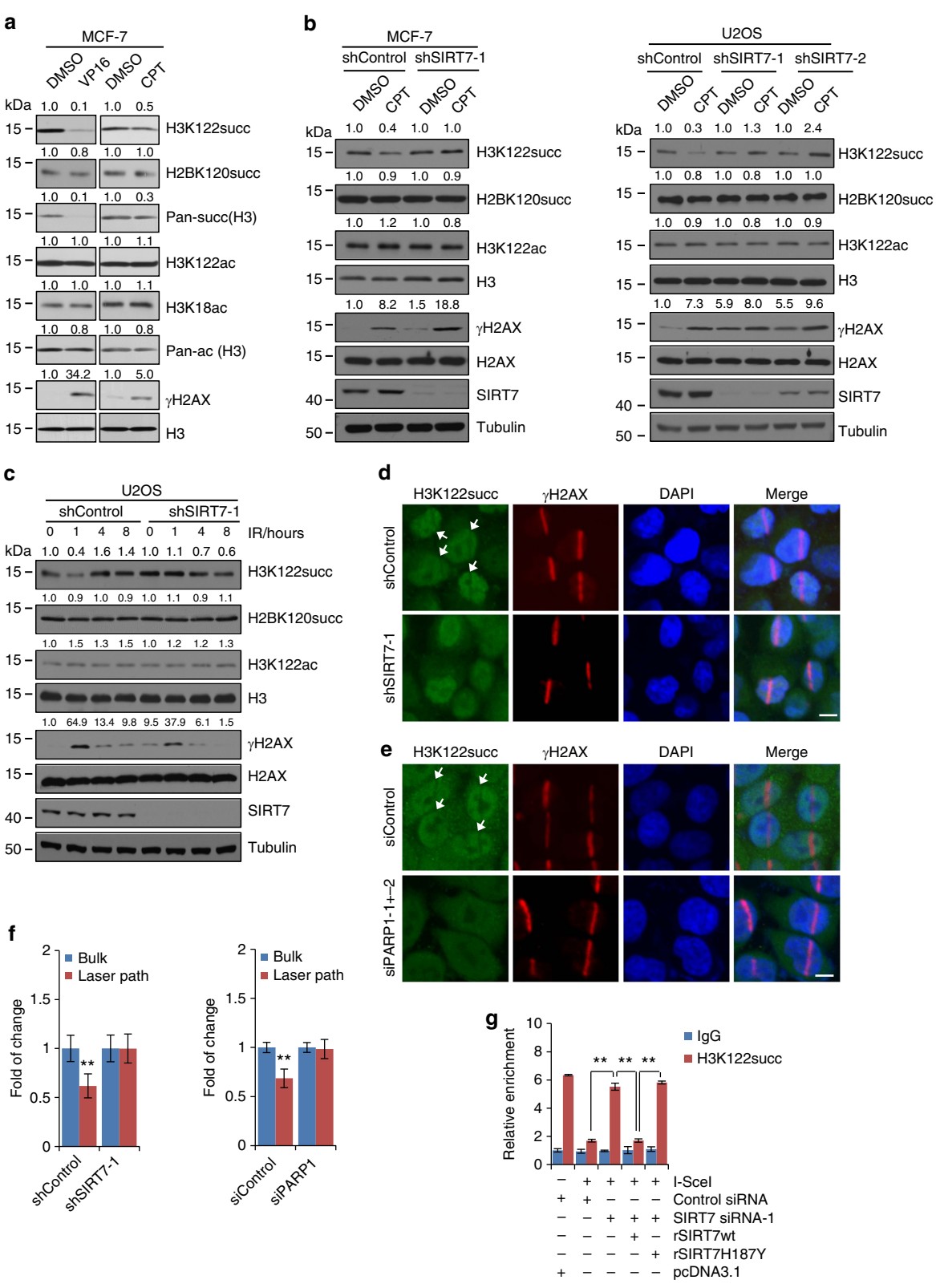

decreases in HR repair efficiency were also detected in DR-GFP-U2OS cells expressing H3K122E or H3K122R (Fig. 7f). These results suggest that both persistent succinylation and non-succinylation of H3K122 could impair HR and NHEJ repair efficiency, suggesting that SIRT7-mediated transient H3K122succ desuccinylation is an integral component of DSB repair process.

### SIRT7-catalysed H3K122 desuccinylation in cell survival.

To explore the biological significance of SIRT7-mediated H3K122succ desuccinylation in DNA repair, we investigated the effect of loss of function of SIRT7 on cell survival and apoptosis in response to DNA-damaging agents. The results showed that KD of SIRT7 in MCF-7 cells resulted in a significant increase in cell apoptosis, and exposure cells to VP16 or CPT severely aggravated this situation (Fig. 8a). Control or SIRT7-depleted MCF-7, U2OS, HCT116, HepG2 cells or SIRT7-knockout U2OS cells were then treated with or without IR at different dosage and subjected to clonogenic survival assays. The results showed that KD or knockout of SIRT7 significantly compromised cell survival in response to IR treatment (Fig. 8b–f). Moreover, clonogenic survival assays demonstrated that U2OS cells stably expressing H3K122E or H3K122R were much more sensitive to IR than cells stably expressing H3wt (Fig. 8g). Collectively, these results suggest that SIRT7-mediated H3K122succ desuccinylation is critically implemented in DNA-damage response and cell survival.

### Discussion

Recent identification of a series of new histone acylations, including succinylation[9,26–29] indicates the epigenetic regulatory circuit is more complicated than we thought. Identification of the writers, readers and erasers of these new histone acylations including succinylation is critically needed for the understanding of the biological significance of these modifications. Neither histone succinylase nor desuccinylase has been described as of today. Although SIRT5 has been reported to be able to regulate succinylation[7], its mitochondrial subcellular localization excludes the possibility for this sirtuin protein to act on nuclear histones.

It is interesting to note that SIRT7S111A mutation shows no obvious effect on desuccinylation of H3K122succ (Fig. 1d and Supplementary Fig. 2a,b) as the corresponding mutations inactivate other members of sirtuin family[32,46,47]. The possible explanations for this discrepancy are discussed as follows: (1) the mutagenesis studies of others' reports focused on different substrates and different enzymatic activities. For example, the mutant of S36 on yeast HST2 (homologues of the SIR2 silencing gene 2) showed significantly reduced $NAD^+$–nicotinamide exchange and deacetylase activity for [$^3$H]-acetylated BSA[32], and loss of ADP-ribosyltransferase activity in another equivalent residue mutant mSIRT6S56A (ref. 46) SIRT7S112A (equivalent to S111A in our study) would abolish the stimulation effect for

pre-rRNA synthesis by wild-type SIRT7 (ref. 47), whereas our study is the first report on the mutagenesis study of SIRT7 desuccinylase activity for histone lysine succinylation; (2) the crystal structures of SIRT2 from *Archaeoglobus fulgidus* (SIR2-Af1)[32] and *Caenorhabditis elegans* (SIRT2)[31] indicate the corresponding residue of SIRT7 S111 lies in the NAD-binding pocket, which is not in direct contact with the observed NAD molecule in the structure[31,32]. In fact, it is buried deep in the active site of the enzyme and does not appear to be structurally important. It is possible that binding of histone succinylated lysine may lead to conformation change of the NAD-binding domain in SIRT7, and hence the position or the conformation of S111 is changed so that it is not directly involved in the desuccinylation reaction.

H3K122 is located on the lateral surface of the histone octamer closing to the dyad symmetry axis[40], the regions with the strongest DNA–histone interaction within the nucleosome[48]. Modifications on nucleosome lateral surface have the potential to facilitate nucleosomal mobilization or histone eviction, thereby modulating chromatin accessibility[49,50]. Moreover, succinyl group differs from acetyl moiety in that succinyl group is two carbons longer and with an additional terminal carboxyl group, potentially generating a stronger stereospecific blockage than acetylation. In addition, compared with acetylation, succinylation is a more acidic modification that disfavours ionic interactions between positively charged lysine side chains and a negatively charged chemical moiety in other molecules, such as DNAs or proteins[30]. It is reasonable to infer that succinylation of the globular histone residues would result in stronger disturbance for the regular nucleotide structure of chromatin than acetylation of the residues on the same site or sites outside the globular domain do, and vice versa. Therefore, succinylation of H3K122 would be potentially associated with a loose chromatin state while desuccinylation of H3K122succ could contribute to the condensation of chromatin structure. Indeed, our results showed that SIRT7-catalysed H3K122succ desuccinylation is linked to chromatin condensation and to efficient DSB repair.

It is documented that chromatin modifiers, including polycomb group proteins[36], HDACs[51], HP1 (ref. 52), nucleosome remodelling and deacetylase complex[36], H3K9me3 methyltransferase SUV39H1 (ref. 53), PRDM2 (ref. 54) and macroH2A1.1 (refs 54,55) that are generally associated with chromatin compaction are also recruited to the sites of damaged DNA, suggesting that chromatin condensation is actively involved in DNA repair process[45,53,54,56]. It is proposed that chromatin condensation is an integral but transient part of the DNA-damage response; condensed chromatin enhances upstream signalling thus promotes DNA repair[45]. Therefore, it is possible that SIRT7-mediated transient H3K122succ desuccinylation at DSB site in the early stage of DNA damage response (DDR) might promote DNA repair by enhancing upstream signalling. It is also suggested that

**Figure 6 | SIRT7 desuccinylates H3K122succ at DSB sites. (a)** MCF-7 cells were treated with 40 nM VP16 or 1 μM CPT for 8 h. Histones were extracted for western blotting analysis with the indicated antibodies. **(b)** Control or SIRT7-depleted MCF-7 cells (left) or U2OS cells (right) were treated with 1 μM CPT for 8 h followed by histone extraction and western blotting analysis with the indicated antibodies. The efficiency of SIRT7 KD and DNA-damage effect induced by CPT were monitored by western blotting of whole-cell lysate using antibodies against SIRT7 and γH2AX, respectively. **(c)** Control or SIRT7-depleted U2OS cells were exposed to 10 Gy of IR and collected at different time points for histone extraction and western blotting analysis with the indicated antibodies. The efficiency of SIRT7 KD and IR treatment was monitored by western blotting of whole-cell lysate using antibodies against SIRT7 and γH2AX, respectively. **(d)** Control or SIRT7-depleted U2OS cells were subjected to laser microirradiation and immunofluorescent analysis of H3K122succ and γH2AX at 5 min after microirradiation. Scale bar, 10 μm. **(e)** U2OS cells transfected with control siRNA or siPARP1 were subjected to laser microirradiation and immunofluorescent analysis of H3K122succ and γH2AX at 5 min after microirradiation. Scale bar, 10 μm. **(f)** H3K122succ levels in **e,f** were measured at and beside damage sites using ImageJ. At least 30 independent cells were scored. Data are represented as mean ± s.e.m. **P < 0.01 (two-tailed Student's t-test). **(g)** Control or DR-GFP-U2OS cells stably expressing rSIRT7wt or rSIRT7H187Y were transfected with control siRNA or siSIRT7-1 as indicated. Twenty four hours later, the cells were transfected with pcDNA3.1 vector or I-SceI for 40 h and subjected to qChIP analysis with antibodies against H3K122succ. Each bar represents the mean ± s.d. for triplicate experiments. **P < 0.01 (two-tailed unpaired Student's t-test).

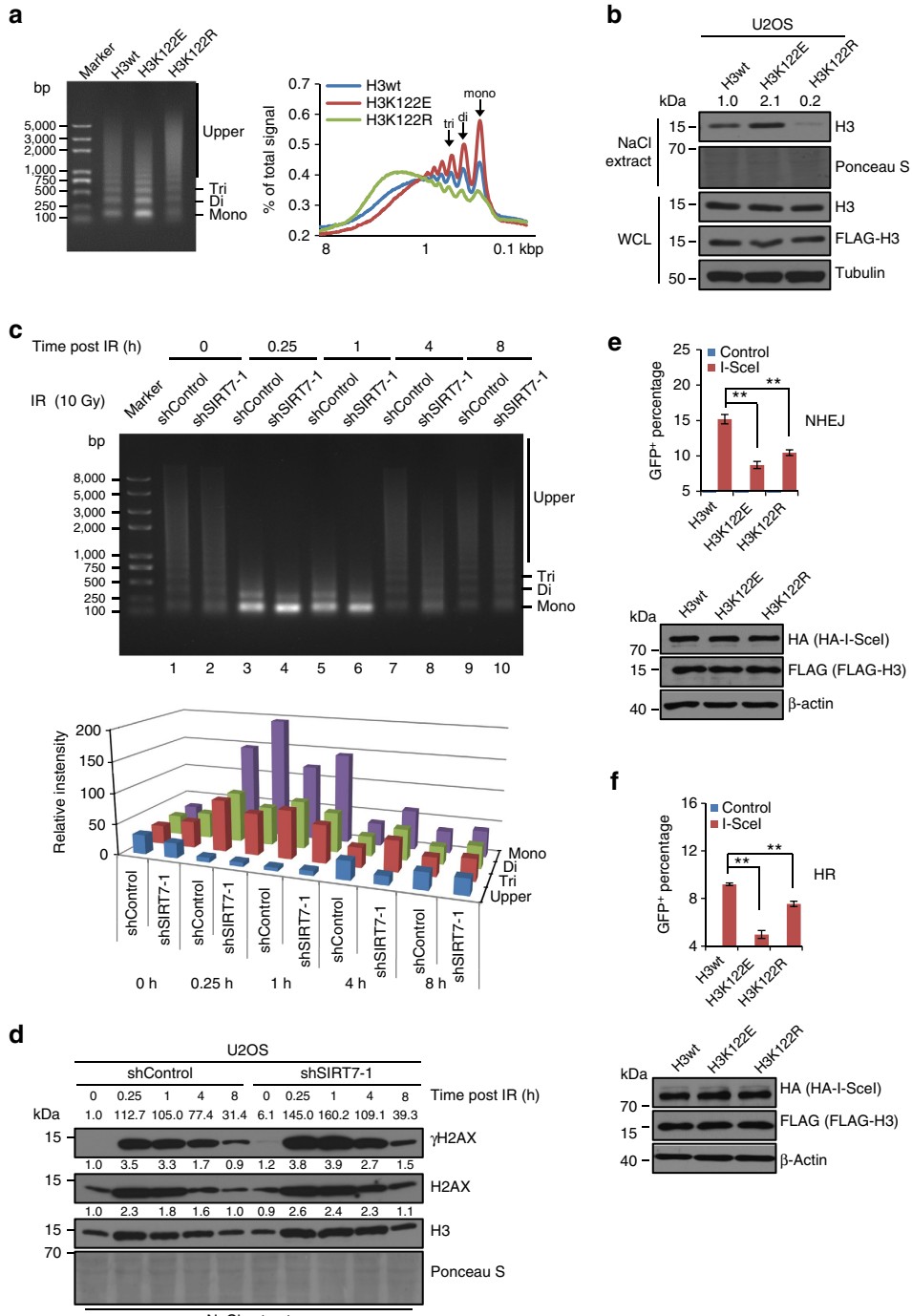

**Figure 7 | SIRT7-catalysed H3K122succ desuccinylation is linked to chromatin condensation during DSB repair.** (**a**) Nuclei from U2OS cells stably expressing FLAG-H3wt, FLAG-H3K122E or FLAG-H3K122R were incubated with 40 gel units of MNase for 5 min followed by DNA extraction and ethidium bromide staining (left). The band densities were quantified using ImageJ software and expressed as percentage of signal minus background of the entire line from top to the bottom. Calibrated kilobase pair (kbp) sizes are indicated (right). (**b**) U2OS cells stably expressing FLAG-H3wt, FLAG-H3K122E or FLAG-H3K122R were extracted in lysate buffer containing 1.5 M NaCl, salt soluble proteins were separated by SDS–polyacrylamide gel electrophoresis (SDS–PAGE), and H3 was detected by western bloting. Ponceau S staining indicates loading. The efficiency of overexpression of FLAG-H3, H3 mutants or total H3 was monitored by western blotting of whole-cell lysate, with corresponding antibodies. (**c**) Control or SIRT7-depleted U2OS cells were exposed to 10 Gy of IR and collected at different time points. Nuclei were prepared and subjected to MNase assays. Mononucleosome, dinucleosome and trinucleosome are indicated (upper). The band densities were quantified using ImageJ software and the intensity values were background subtracted (lower). (**d**) Control or SIRT7-depleted U2OS cells were exposed to 10 Gy of IR and collected at different time points. Cells were extracted in lysate buffer containing 1.0 M NaCl, salt-soluble proteins were separated by SDS–PAGE, and γH2AX, H2AX and H3 were detected by western bloting. Ponceau S staining indicated loading. (**e**,**f**) Overexpression of H3K122 mutants affected the repair efficiency of NHEJ and HR. EJ5-GFP-HEK293 (**e**) or DR-GFP-U2OS (**f**) cells stably expressing FLAG-H3wt, FLAG-H3K122E or FLAG-H3K122R were transfected with I-SceI for 48 h and analysed by FACS. Each bar represents the mean ± s.d. for triplicate experiments. **P < 0.01 (two-tailed unpaired Student's t-test). The efficiency of overexpression of FLAG-H3, H3 mutants or HA-I-SceI was monitored by western blotting.

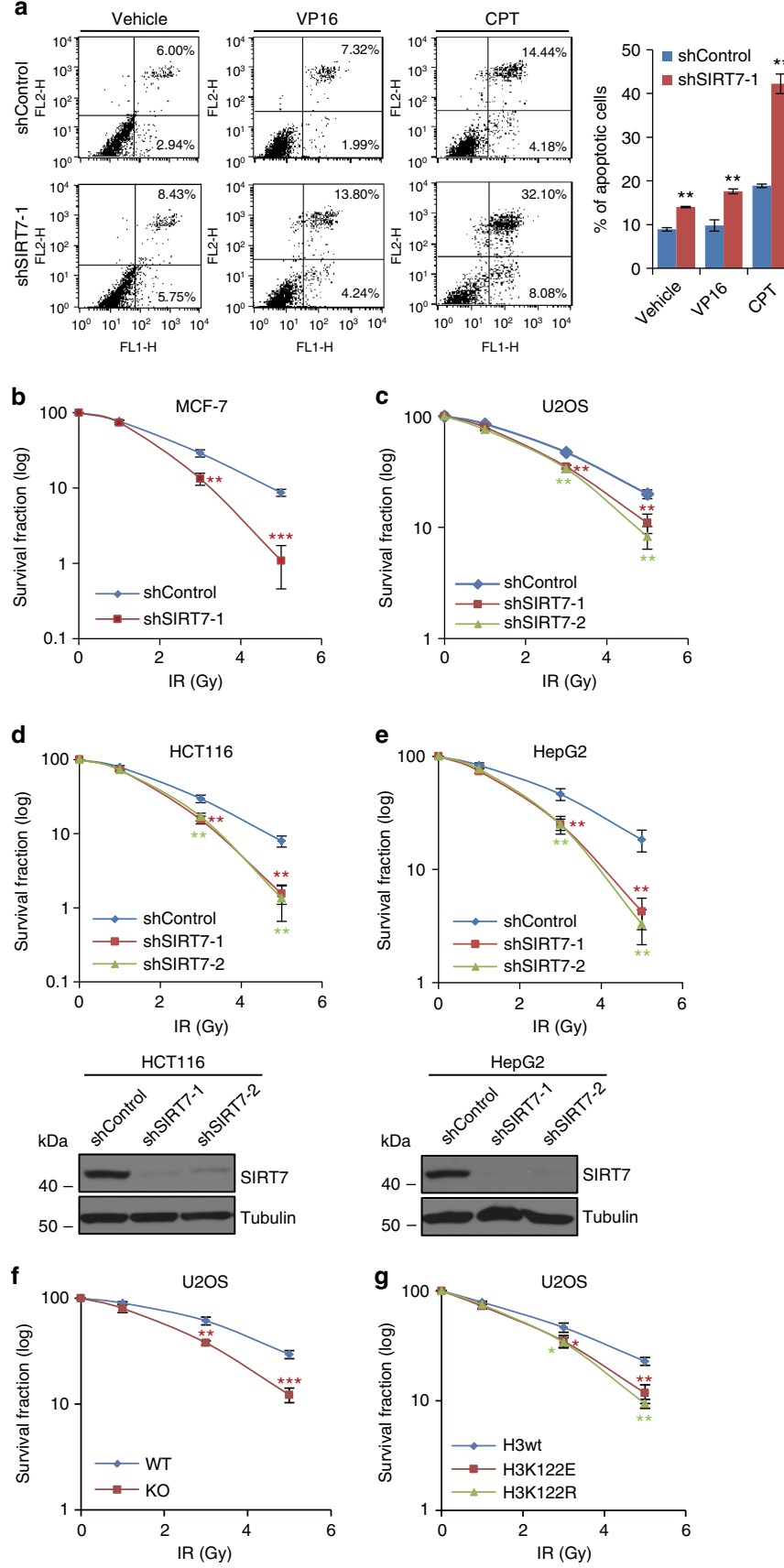

condensation of chromatin is required for inhibiting local transcription, compacting the local chromatin structure, rewriting the local epigenetic landscape, and limiting DSB mobility during the initial moments following DSB production[53]. Clearly, the biological significance of SIRT7-associated chromatin condensation needs further elucidation.

Recent reports indicate that PAR-dependent accumulation of transcription repression-associated chromatin regulators such as histone variant macroH2A1.1 (ref. 55), the nucleosome remodelling and deacetylase complex[36], ALC1 (ref. 57) or SUV39H1 (ref. 53) function to modulate chromatin structures at sites of DNA breaks to facilitate signalling and/or repair of DNA damage. Interestingly, both PARP1 and SIRT7 are enzymes requiring $NAD^+$ as coenzyme, and, intriguingly, several studies indicate that $NAD^+$-consuming enzymes such as PARPs or cADP-ribose synthase influence sirtuin activity by restricting $NAD^+$ availability[58]. For example, it is reported that PARP1 activation was associated with a depletion of $NAD^+$ pool thus an inhibition of SIRT1 activity, leading to the death of cells[59,60]. It is believed that PARP catalysis is the main $NAD^+$ catabolic source in cells that forces the cell to continuously synthesize $NAD^+$ from the *de novo* pathway or recycling pathway in the case of cellular stresses, especially during the DNA repair process[61]. Whether or not the PARP1-dependent recruitment of SIRT7 during DNA-damage response is a reflection of coordinated enzymatic actions between PARP1 and SIRT7 in terms of $NAD^+$ usage is currently unknown. In light of the reports that several other sirtuins including SIRT1 (ref. 62) and SIRT6 (refs 63,64) are also involved in DNA-damage repair, it will be interesting in future investigations to investigate the importance of metabolic processes in chromatin remodelling during DNA repair. In addition, further studies are needed to elucidate the full spectrum of the regulation of histone desuccinylation and to decipher the histone languages encoded by this modification. Moreover, due to technical limitations, our current study focuses on H3K122; regulation of desuccinylation by SIRT7 on other histone sites cannot be excluded. It is also possible that the reported SIRT7 substrates and SIRT7-associated desuccinylation coordinately influence chromatin environment during DNA-damage repair. Nevertheless, our study indicates that SIRT7 is a $NAD^+$-dependent histone desuccinylase, providing a molecular basis for the understanding of epigenetic regulation by this sirtuin protein. Our experiments revealed that SIRT7-catalysed H3K122 desuccinylation is critically implemented in DNA-damage response and cell survival, providing a mechanistic insight into the cellular function of SIRT7.

## Methods

**Plasmids.** The cDNA for wild-type SIRT5, SIRT6 or SIRT7 was amplified by PCR and ligated into pcDNA3.1( − ) plasmid containing a FLAG tag. SIRT7 mutants including S111A, H187Y and S111A/H187Y were generated by using QuikChange Lightning Site-Directed Mutagenesis Kit. The GFP-SIRT7 was constructed by cloning full-length of SIRT7 into pEGFP-N1 vector. SIRT7 siRNA-1 resistant pcDNA3.1( − )-FLAG-SIRT7wt (rSIRT7wt) and SIRT7 siRNA-1 resistant pcDNA3.1( − )-FLAG-SIRT7H187Y (rSIRT7H187Y) were generated by synonymous mutations (G606A, G609A and C612T). The pLVX-IRES-FLAG-rSIRT7wt and pLVX-IRES-FLAG-rSIRT7H187Y for lentiviral production were subcloned from pcDNA3.1( − )-FLAG-rSIRT7wt/H187Y. The PITA-FLAG-H3wt for lentiviral production was subcloned with a C-terminal FLAG tag from pBOS-HA-H3.1. PITA-FLAG-H3wt was used as a template to obtain the PITA-FLAG-H3K122E and PITA-FLAG-H3K122R mutants by standard site-directed mutagenesis. All clones were confirmed by DNA sequencing.

**Antibodies and reagents.** The polyclonal antibodies against H3K122succ and H2BK120succ were generated by immunizing rabbits with a synthetic succinyl peptide corresponding to residues surrounding K122 of human histone H3 or K120 of human histone H2B. Antibodies were purified by protein A-conjugated agarose followed by affinity chromatography with K122 succinylated histone H3 or K120 succinylated histone H2B peptides. The sources of the other antibodies against the following proteins were as follows: FLAG (F3165), α-tubulin (clone B-5-1-2, T6074) and β-actin (A1978) from Sigma; SIRT7 (sc-365344), Ku80 (sc-5280) and PARP1/2 (sc-7150) from Santa Cruz Biotechnology; γH2AX from Millipore (05-636) and Cell Signaling Technology (9718P); H3K122ac (ab33309), H3 (ab1791), H2AX (ab11175) and BrdU (ab8039) from Abcam; H3K18ac (PTM-114), pan-succinylation (PTM-401) and pan-acetylation (PTM-105) from PTM BioLabs; BRCA1 (22362-1-AP) from Proteintech; haemagglutinin (HA) (M180-3) from MBL; SIRT6 from Abgent (AP-6245a); and agarose beads conjugated with pan anti-succinyllysine (PTM-402), crotonyllysine (PTM-503) and malonyllysine (PTM-904) antibodies were purchased from PTM BioLabs.

VP16 (E1383), camptothecin (C9911), anti-FLAG M2 affinity gel (A2220), $1 \times$ FLAG peptide (F3290), PJ-34 (P4365) and BrdU (B5002) were from Sigma. KU-55933 (118500) and $NAD^+$ (20-221) were from Millipore. NAM (N814605) was from Macklin. Protein A/G Sepharose CL-4B beads were from Amersham Biosciences, and protease inhibitor mixture cocktail was from Roche Applied Science.

**Cell culture and transfection.** MCF-7, U2OS, HeLa, HepG2, HCT116 and HEK293T cells were from the American Type Culture Collection. DR-GFP-U2OS and EJ5-GFP-HEK293 cell lines were kindly provided by Dr Xingzhi Xu (Capital Normal University, Beijing). The cells were maintained in Dulbecco's modified Eagle's medium supplemented with 10% fetal bovine serum (FBS). Transfections were carried out using Lipofectamine 2000 (Invitrogen) according to the manufacturer's instructions. The sequences of siRNAs are given in Supplementary Table 1. SIRT7 siRNAs were synthesized by Sigma. Ku80 and BRCA1 siRNAs were synthesized by Suzhou GenePharma. siRNA oligonucleotides were transfected into cells using RNAiMAX (Invitrogen) according to the manufacturer's instructions.

**Lentiviral production and infection.** RNAi lentivirus system was constructed using pLKO.1 according to protocols described online (http://www.addgene.org/tools/protocols/plko/#E). The sequences of short hairpin RNAs are given in Supplementary Table 2. In brief, short hairpin RNA sequences targeting human SIRT7 (TRCN0000359594, shSIRT7-1; TRCN0000359663, shSIRT7-2) or SIRT6 (TRCN0000232532) were cloned into pLKO.1. The recombinant constructs, as well as assistant vectors psPAX2 and pMD2.G, were co-transfected into HEK293T cells. Viral supernatants were collected 48 h later, clarified by filtration through 0.45-µm filters and concentrated by ultracentrifugation. The concentrated viruses were used to infect $5 \times 10^5$ cells (20–30% confluent) in a 60-mm dish with 8 µg ml$^{-1}$ polybrene. Infected cells were selected with 1.5 µg ml$^{-1}$ puromycin (Amresco). The lentivirus carrying rSIRT7wt, rSIRT7H187Y, FLAG-H3.1wt, FLAG-H3.1K122E and FLAG-H3.1K122R were packaged and collected similarly.

**SILAC labelling and quantitative proteomics analysis.** Control or SIRT7 KD MCF-7 cells were grown in Dulbecco's modified Eagle's medium supplemented with 10% FBS and either the 'heavy' form of [U-$^{13}$C$_6$]-L-lysine or 'light' [U-$^{12}$C$_6$]-L-lysine for more than six generations before being collected to achieve more than 97% labelling efficiency. After that, the cells were further expanded in SILAC media to desired cell number ($\sim 5 \times 10^8$) in $15 \times 150$-mm$^2$ flasks. The cells were then collected and the core histones were isolated and digested. Lysine crotonylation (Kcro), succinylation (Ksucc) and malonylation (Kmal) peptides were then enriched by pre-washed antibody beads (PTM Biolabs, Hangzhou). The eluted peptides were cleaned with C18 ZipTips (Millipore) according to the manufacturer's instructions, followed by analysis with LC–MS/MS. The resulting MS/MS data were processed by using MaxQuant with integrated Andromeda search engine (version 1.4.1.2). False discovery rate thresholds for protein, peptide and modification site were specified at 1%.

**Figure 8 | SIRT7-mediated H3K122succ desuccinylation is implemented in cellular response to DNA damage.** (**a**) Control or SIRT7-depleted MCF-7 cells were treated with 40 nM VP-16 or 1 µM CPT, and collected for annexin V and propidium iodide double staining. Cell apoptosis was determined by flow cytometry. Data were represented as mean ± s.d. (**b**–**e**) Control or SIRT7-depleted MCF-7 (**b**), U2OS (**c**), HCT116 (**d**) or HepG2 (**e**) cells were treated with or without IR at the indicated doses and then subjected to clonogenic survival assays. The efficiency of SIRT7 KD in HCT116 (**d**) or HepG2 (**e**) cells was monitored by western blotting of whole-cell lysate, with corresponding antibodies. (**f**) Control or SIRT7-knockout U2OS cells were treated with or without IR at the indicated doses and then subjected to clonogenic survival assays. (**g**) U2OS cells stably expressing FLAG-H3wt, FLAG-H3K122E or FLAG-H3K122R were treated with or without IR at the indicated doses and then subjected to clonogenic survival assays. Data were represented as mean ± s.d. for triplicate experiments. *$P < 0.05$, **$P < 0.01$ and ***$P < 0.001$ (two-tailed unpaired Student's *t*-test).

**Western blotting.** Western blotting was performed according to standard procedures. Antibodies used were anti-SIRT7 (Santa Cruz Biotechnology, sc-365344, 1:500), anti-HA (MBL, M180-3, 1:2,000), anti-Flag (Sigma, F3165, 1:10,000), anti-β-actin (Sigma, A1978, 1:10,000), anti-tubulin (Sigma, clone B-5-1-2, T6074, 1:50,000), anti-SIRT6 (Abgent, AP-6245a, 1:500), anti-PARP1/2 (Santa Cruz Biotechnology, sc-7150, 1:5,000), anti-Ku80 (Santa Cruz Biotechnology, sc-5280, 1:2,000), anti-BRCA1 (Proteintech, 22362-1-AP, 1:1,000), anti-γH2AX (Millipore, 05-636, 1:2,000), anti-H2AX (Abcam, ab11175, 1:2,000) anti-pan-acetylation (PTM BioLabs, PTM-105, 1:1,000), anti-pan-succinylation (PTM BioLabs, PTM-401, 1:1,000), anti-H3K122succ (1:4,000), anti-H2BK120succ (1:8,000), anti-H3K122ac (Abcam, ab33309, 1:2,000), anti-H3K18ac (PTM BioLabs, PTM-114, 1:1,000), anti-H3 (Abcam, ab1791, 1:100,000) and anti-rabbit (Jackson ImmunoResearch, 115-035-003, 1:8,000) or anti-mouse (Jackson ImmunoResearch, 111-035-003, 1:8,000) secondary antibodies conjugated to horseradish peroxidase. The bands were quantified by densitometry with ImageJ software. Uncropped scans of the most important blots are shown in Supplementary Fig. 11.

**Protein purification.** Protein purification was performed as described previously[65] with some optimization. Briefly, for FLAG-SIRT7wt, FLAG-SIRT7H187Y, FLAG-SIRT5 and FLAG-SIRT6, HEK293T cells expressing full-length FLAG-tagged SIRT7wt, SIRT7H187Y, SIRT5 or SIRT6 were collected and lysed in lysis buffer (50 mM Tris·HCl (pH 7.4), 300 mM NaCl, 1% Nonidet P-40, 1 mM EDTA, 10% (vol/vol) glycerol and 1 mM dithiothreitol (DTT)) supplemented with protease inhibitors (Roche). The resulting lysate was incubated with anti-FLAG M2 affinity gel for 2 h and the beads were washed five times with lysis buffer. The immobilized proteins was eluted with $1 \times$ FLAG peptide and used in desuccinylation assays as described below or resolved on SDS–PAGE followed by Coomassie brilliant blue staining.

**Preparation of mononucleosome.** Preparation of mononucleosomes was conducted according to the procedure described previously[66]. Briefly, HeLa cells were collected by ice-cold PBS, resuspended in lysis buffer (10 mM Tris·HCl (pH 7.5), 10 mM NaCl, 3 mM MgCl$_2$ and 0.4% Nonidet P-40) in the presence of protease inhibitors and the nuclei were pelleted. Glycerol buffer (10 mM Tris·HCl (pH 7.4), 0.1 mM EDTA, 5 mM MgAc$_2$ and 25% (vol/vol) glycerol) was add to get a final concentration of 1 to 2 mg ml$^{-1}$ nuclei. To generate nucleosomal material, digestions were conducted by adding 1 volume of $2 \times$ MNase buffer (50 mM KCl, 8 mM MgCl$_2$, 2 mM CaCl$_2$ and 100 mM Tris·HCl (pH 7.4)) and 3,000–8,000 gel units per ml MNase. The reaction was incubated for 15 min at 37 °C and stopped by adding EDTA to a final concentration of 10 mM. The mononucleosomes were then purified by sucrose gradient assay.

**Histone desuccinylation assay.** The sequence of synthesized H3K122 (117-128) succinyl peptide was IRRYQK(succinyl)STELLI. The identity and purity of the peptides were verified by LC–MS. Two micrograms of purified FLAG-SIRT7wt, FLAG-SIRT7H187Y, FLAG-SIRT5 or FLAG-SIRT6 were incubated with 500 ng H3K122succ peptides in desuccinylation assay buffer[8] (20 mM Tris·HCl (pH 7.5) and 1 mM DTT) with or without 1.0 mM NAD$^+$ in a final volume of 30 μl for 2 h at 37 °C. The reaction mixture was boiled and subjected to dot blot analysis. One microgram of calf thymus bulk histones (Sigma) or mononucleosomes isolated from HeLa cells were incubated with 0.25–5 μg of SIRT7wt, SIRT7H187Y, SIRT5 or SIRT6 in desuccinylation assay buffer in the presence or absence of 1.0 mM NAD$^+$ and/or 10 mM NAM in a final volume of 30 μl for 2 h at 37 °C. The reaction mixture was boiled in SDS sample buffer and subjected to SDS–PAGE analysis and mass spectra analysis.

**Immunopurification and mass spectrometry.** HEK293T cells transfected with empty vector or FLAG-SIRT7wt for 48 h were lysed in lysis buffer (50 mM Tris·HCl (pH 7.4), 150 mM NaCl, 0.3% Nonidet P-40, 1 mM DTT and 5 mM EDTA) plus protease inhibitors (Roche) for 30 min at 4 °C. This was followed by centrifugation at 14,000g for 15 min at 4 °C. Protein supernatant was incubated with anti-FLAG M2 gel for 2 h at 4 °C. After washing with lysis buffer for five times, $1 \times$ FLAG peptide was used to elute the protein complex from the beads following the manufacturer's instructions. The eluted protein complex was then resolved on NuPAGE 4–12% Bis-Tris gel (Invitrogen), silver stained and subjected to LC–MS/MS for sequencing and data analysis.

**Laser microirradiation and X-ray irradiation.** For time-lapse imaging of living cells, cells grown on a dish with thin glass bottom (NEST) in the presence of 10 μM of 5-bromo-2′-deoxyuridine (BrdU, Sigma-Aldrich) in phenol red-free medium (Invitrogen) for 24 h were locally irradiated with a 365-nm pulsed nitrogen ultraviolet laser (16 Hz pulse, 45% laser output) generated from the micropoint system (Andor). This system was directly coupled to the epifluorescence path of the Nikon A1 confocal imaging system with time-lapse imaging every 30 s for 15 min. A heated stage with an objective lens heater was used to keep the cells at the appropriate temperature (37 °C) and growth conditions during imaging. Images were analysed using ImageJ software. For quantification of protein accumulations

at laser-generated DSBs, the mean fluorescence intensity within the regions of interest (ROI) was measured for each time point. The intensity values were background subtracted, and the ratio of intensity within the microirradiated nuclear area to non-microirradiated area was calculated. At least 30 independent cells were scored. For laser microirradiation and immunofluorescence assays, cells were grown on LabTek II chamber slides (Thermo Scientific) in the presence of 10 μM BrdU in phenol red-free medium (Invitrogen) for 24 h before induction of DNA damage by a ultraviolet-A laser ($\lambda = 355$ nm, 40% energy) using a Zeiss Observer.Z1 inverted microscope with a PALM MicroBeam laser microdissection workstation. After irradiation, the cells were incubated at 37 °C for an appropriate time and processed for immunostaining. IR was delivered by an X-ray generator (RS2000 PRO, 160 kV, 25 mA; Radsource Corporation).

**Immunofluorescence.** Cells were washed with PBS, fixed in 4% paraformaldehyde for 10 min. Specifically, for H3K122succ stain, before fixed in 4% paraformaldehyde, cells were washed once with cold PBS, extracted with CSK buffer (10 mM Pipes (pH 7.0), 100 mM NaCl, 300 mM sucrose, 3 mM MgCl$_2$ and 0.5% Triton X-100) for 2 min, washed again with cold PBS. Then the cells were permeabilized with 0.2% Triton X-100 and incubated with appropriate primary antibodies (SIRT7, Santa Cruz Biotechnology, sc-365344, 1:100; γH2AX, CST, 9718P, 1:400; H3K122succ, 1:100; γH2AX, Millipore, 05-636, 1:100) and secondary antibodies coupled to Alexa Fluor 488 (Jackson ImmunoResearch, rabbit, 111-545-003, 1:100) or Alexa Fluor 594 (Jackson ImmunoResearch, mouse, 115-585-003; rabbit, 111-585-003, 1:100). The cells then were washed for four times, and a final concentration of 0.1 μg ml$^{-1}$ 4,6-diamidino-2-phenylindole dihydrochloride (Sigma) was included in the final wash to stain nuclei. Images were acquired with a FluoView FV1000 laser scanning confocal system (Olympus) connected to an inverted microscope (IX-81) equipped with PLAPON $\times$ 60 oil/numerical aperture 1.42 objective. To avoid bleed-through effects in double-staining experiments, each dye was scanned independently in a multi-tracking mode.

**Chromatin immunoprecipitation.** ChIP experiments were performed according to the procedure described previously[67]. About 10 million cells were crosslinked with 1% formaldehyde for 10 min at room temperature and quenched by the addition of glycine to a final concentration of 125 mM for 5 min. The fixed cells were resuspended in SDS lysis buffer (1% SDS, 5 mM EDTA and 50 mM Tris·HCl (pH 8.1)) in the presence of protease inhibitors and 10 mM NAM, then subjected to $3 \times 10$ cycles (30 s on and off) of sonication (Bioruptor, Diagenode) to generate chromatin fragments of ~300 bp in length. Lysates were diluted in buffer containing 1% Triton X-100, 2 mM EDTA, 20 mM Tris·HCl (pH 8.1), 150 mM NaCl plus 10 mM NAM and protease inhibitors. For immunoprecipitation, the diluted chromatin was incubated with control or specific antibodies (3–5 μg) for 12 h at 4 °C with constant rotation, and 50 μl of 50% (vol/vol) protein A/G Sepharose beads was then added and the incubation was continued for an additional 2 h. Beads were washed with the following buffers: TSE I (0.1% SDS, 1% Triton X-100, 2 mM EDTA, 20 mM Tris·HCl (pH 8.1) and 150 mM NaCl); TSE II (0.1% SDS, 1% Triton X-100, 2 mM EDTA, 20 mM Tris·HCl (pH 8.1) and 500 mM NaCl); buffer III (0.25 M LiCl, 1% Nonidet P-40, 1% sodium deoxycholate, 1 mM EDTA and 10 mM Tris·HCl (pH 8.1)); and Tris-EDTA buffer. Between washes, the beads were collected by centrifugation at 4 °C. The pulled-down chromatin complex together with input was de-crosslinked at 70 °C for 2 h in elution buffer (1% SDS, 5 mM EDTA, 20 mM Tris·HCl (pH 8.1), 50 mM NaCl and 0.1 mg ml$^{-1}$ proteinase K). Eluted DNA was purified with PCR purification kit (Qiagen) and analysed by quantitative PCR using primers described in Supplementary Table 3.

**Nucleosome stability assay.** Nucleosome stability assays were performed as described previously[44]. Briefly, cells were collected and washed twice in ice-cold PBS by centrifugation at 500g. Cell pellet was resuspended completely in 500 μl buffer A (20 mM HEPES (pH 7.9), 0.5 mM DTT, 1 mM phenylmethyl sulphonyl fluoride, 1.5 mM MgCl$_2$ and 0.1% Triton) containing 1.0 or 1.5 M NaCl. Cells were incubated for 40 min at 4 °C with constant agitation. Samples were then centrifuged at 100,000g (Ultracentrifuge; HITACHI) for 20 min, and the supernatant, containing released histones, retained for further analysis.

**Generation of SIRT7 knockout cell lines by CRISPR-Cas9.** Three single-guide RNAs (sgRNAs 1–3) that target different regions of the human SIRT7 gene were selected from previously published genome-wide human sgRNA Libraries[68]. The sequences are given in Supplementary Table 4. Oligos corresponding to the sgRNAs were cloned into the GV392 vector containing the *hSPCas9* gene and a puromycin selection marker gene. U2OS cells were transfected with either of the sgRNAs and selected with puromycin 48 h post transfection. Single clones were retrieved after 7 days of puromycin selection, expanded and analysed for abrogation of SIRT7 expression by western blotting. Manifestation of the SIRT7 mutations was verified by PCR and sequencing.

**Cell flow cytometry.** For measurement of repair efficiency, DR-GFP-U2OS or EJ5-HEK293 cells were trypsinized, washed with PBS and collected with

FACSCalibur. The data were analysed by FlowJo. For analysis of BrdU incorporation, cells were pulsed with 10 µM BrdU for 15 min followed by trypsinization, PBS wash and fixation with ice-cold 70% ethanol. Cells were then resupended in 2 N HCl, 0.5% Triton X-100 and incubated for 30 min at room temperature. Next, cells were resuspended in 0.1 M $Na_2B_4O_7$ (pH 8.5), spin down and resuspended in antibody incubation buffer (1% BSA and 0.5% Tween 20 in PBS) containing anti-BrdU (Abcam, ab8039, 1:1,000) for 30 min before the addition of the secondary antibody (Alexa Fluor 488-conjugated goat anti-mouse IgG (H = L), Jackson ImmunoResearch, 115-545-003, 1:200) for 30 min. Finally, cells were washed and incubated in propidium iodide buffer and analysed by FACS. Acquisition of the data is performed through Cell Quest software, and analysis through FlowJo and ModFit. Apoptosis was measured using Annexin V-FITC Apoptosis Detection Kit (BD Pharmingen) according to the manufacturer's instructions and analysed using the FACSCalibur flow cytometer.

**MNase sensitivity assay.** About 1 million cells were washed with cold PBS, resuspended in ice-cold Nonidet P-40 cell lysis buffer (10 mM Tris · HCl (pH 7.5), 10 mM NaCl, 3 mM $MgCl_2$ and 0.4% Nonidet P-40) in the presence of protease inhibitors and incubated on ice for 5 min, The lysate was cleared with centrifugation at 2,000g for 5 min at 4 °C. The resulting pellet was collected and washed with lysis buffer twice. The pellet was then resuspended in 50 µl glycerol buffer (10 mM Tris · HCl (pH 7.4), 0.1 mM EDTA, 5 mM MgAc$_2$ and 25% (vol/vol) glycerol), mixed with equal volume of 2 × MNase buffer (50 mM KCl, 8 mM $MgCl_2$, 2 mM $CaCl_2$ and 100 mM Tris · HCl, (pH 7.4)), and incubated at 37 °C for 5 min with MNase (NEB) at the indicated amount per 100 µl of total reaction volume. The reaction was stopped by adding EDTA at the final concentration of 10 mM. Genomic DNA was purified and separated by electrophoresis in 1.2% agarose gel.

**Clonogenic survival assay.** Cells were plated in 12-well plates in triplicates (400 cells per well) and were subsequently treated with IR and let to grow in colonies for 10 days. After 10 days, the cells were washed with PBS, fixed with 4% formaldehyde for 10 min and stained with crystal violet (0.1% wt/vol) for 20 min. The number of colonies per well was counted, and the plating efficiency and surviving fraction for given treatments were calculated on the basis of the survival rates of nonirradiated cells.

**Statistical analysis.** The data were analysed by a two-tailed unpaired Student's t-test (GraphPad Prism software, version 5.01) and expressed as mean ± s.d. unless otherwise indicated. $P < 0.05$ was considered to be statistically significant.

**Data availability.** All data presented is presented in this manuscript or available from the authors on request.

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

## Acknowledgements

This work was supported by grants 81071677 (to W.Y.), and 91219201, 81530073 and 81130048 (to Y.S.) from the National Natural Science Foundation of China, and a grant (2014CB542004 to J.L.) from the Ministry of Science and Technology of China.

## Author contributions

L.L., L.S., W.Y. and Y.S. conceived the project and designed the experiments; L.L., Lan S., W.Y., S.Y., D.Z. and R.Y. performed and interpreted molecular and cell biology experiments; L.L., L.S., W.Y. and Y.S., wrote and revised the manuscript. C.Z., J.Y., L.H. and W.J. designed and performed the quantitative mass spectrometry in Fig. 1c; L.L., L.S., X.Y., LY.S., J.L., Lei S., Y.S. and W.Y. analysed the data and provided technical assistance; L.L. and L.S. made independent contributions to the work.

## Additional information

**Competing financial interests:** The authors declare no competing financial interests.

