## [Peer Review File · Nature Communications]

Reviewers' comments:

Reviewer #1 (Remarks to the Author):

In this manuscript, Yu and colleagues propose that the mammalian SIRT7 protein is an histone desuccinylase, and such activity is critical for SIRT7 function in PARP-1 dependent DNA repair. Histone succinylation has been previously described, yet the specific enzymes regulating this modification, as well as its functional roles in mammals remain poorly known. Furthermore, SIRT7 remains one of the least characterized sirtuins, and as such, describing a novel role for this enzyme in DNA repair is highly significant. These studies, if conclusive, provide novel insights that could impact the fields of epigenetics, sirtuins and DNA repair, and as such, worthy of publication in this journal. However, many of the experiments presented, in particular those related to the novel desuccinylase activity, fall short in supporting the authors' hypotheses, as indicated in detail below, and they are, at best, preliminary.

Major comments:

- The rationale to use only MCF7 and 293T for the biochemical assays is not clear. Other lines exhibit equally "low" or "high" levels, and since the authors do not validate their experiments in more physiological conditions (in vivo), their results will be more convincing if replicated in other lines. In this context, the study will benefit if some of the claims made in knockdown cells will be shown in SIRT7 KO cells.
- Although the mass spectrometry assays seem solid, the biochemistry to support these results is not convincing. For instance, most of the westerns presented exhibit modest changes in the putative succinylation sites. It is well established that histone modifications are difficult to quantify in westerns, given intrinsic variability in signal. All these westerns should be performed at least in triplicates, and densitometry should be performed in order to provide a quantification of these experiments. As it stands, the data does not look convincing. Particularly, Fig. 1D, it seems the line for the Pan-succ antibody has a bubble exactly for the Sirt7wt and SIRT7S11A extracts. Similarly, their newly generated antibodies should be further characterized. In Fig. 1F, the authors show that the presence of succ. peptides in competition assays completely eliminated the signal in the westerns, yet they did not show other peptides (with different modification, or even H3 unmodified peptides) as negative controls. The changes claimed in Fig. 1G are modest. Without quantification and replicates, these results are inconclusive. The same could be said for all the blots in Fig. 2C-D, and those in Fig. 4H-I.
- In this context, the western blot in Fig. 1H is worrisome. The authors claim that only NAM treatment increased H3K122succ levels, yet it seems that also TSA and Sodium butyrate

increased its levels, a result that goes against their hypothesis. Of concern, treatment with TSA should majorly increase pan-acetylation, yet the western showed only a modest effect (despite the authors claiming "marked increase"). Such results raise concerns on the solidity of the whole western blot.

- The IF shown in Fig. 1E is not convincing. Has this antibody been tested in IF assays before. The authors should include controls to verify the antibody truly recognize histone succinylation.

- Loading controls and quantification should be included in the dot blot assays in Fig. 2B.

- The data in Fig. 3, indicating recruitment of SIRT7 to sites of DSBs is compelling.

- The experiments shown in Fig.4, demonstrating a role for SIRT7 in both HR and NEHJ DNA repair seem robust. It will be reassuring to see the original FACs plots for at least some of these experiments.

- However, in this context, how long after ISceI transfection was the experiment in figure 4J done? It is very difficult to assess any kinetics in this setting since this system is based on cut-repair cycles in the presence of the ISceI enzyme.

- Although the recruitment of SIRT7 to sites of breaks is clear, the mechanistic roles for such recruitment are less clear. In this context, the experiments provided to claim a role in modulating chromatin compaction are much less convincing. MNase experiments are known to have high variability and, most of the times, is very difficult to get informative results. To overcome this, it is imperative to repeat the experiment several times using different amounts of MNase and times of digestion to see a pattern of more digested chromatin with increasing units of enzyme or time of digestion (as a control that the experiment is working). It is surprising then that the authors conclude that H3K122succ and Sirt7 regulate chromatin condensation by doing just one experiment, in which this reviewer cannot see any significant difference. Moreover, if H3K122succ induces chromatin relaxation and this mark decreases after 1h of IR (fig 4I), why is chromatin in a more open state (fig 5b)?

- Assuming that SIRT7-dependent desuccinylation indeed enhances chromatin compaction, how such compaction will benefit DNA repair? One would think that increased compaction may decrease recruitment of repair factors, in turn negatively influencing DNA repair, which is the opposite to what the authors claim.

- In Fig. 5C, why do the control "H3wt" cells exhibit 15% GFP cells, while in the previous Fig. 4A, control cells showed around 8-10% GFP? Such discrepancy in control assays raise concerns about reproducibility, in particular when cells expressing the transfecting mutant histones exhibit ~8-10% GFP, which are levels of control cells in the previous experiment.

- In Figure 6 B-D: the DNA damage sensitivity assays show a very modest effect (please note the scale is set up to enhance the difference. Survival assays should be plotted in log scale). P-values

should be indicated to determine whether the difference claimed is significant. Can these experiments be repeated in other lines?

Minor concerns:

- Few grammatical errors should be corrected. For instance, on page 4 "in last decade" should be "in the last decade". On page 20, "a residue could not be succinylated" should be "a residue that could not be succinylated".

Reviewer #2 (Remarks to the Author):

In this study, Shi et al report a new enzymatic activity of the histone/protein deacetylase SIRT7, de-succinylation of histone H3 at Lysine K122. They show that SIRT7 associates with chromatin at DNA double-strand breaks (DSBs), where its desuccinylation of H3K122 is important for chromatin compaction and DSB repair. The identification of new enzymatic activities and substrates of SIRT7 is important for the SIRT7 field, and the identification of the functions and regulatory enzymes of new histone modifications such as succinylation are important for the chromatin field. Overall the data are clear and the experiments are well described.

Major concerns:

1. In Figure 4H and I, SIRT7 RNAi seems to increase baseline γ -H2AX but does not increase γ -H2AX upon DNA damage (in fact, in 4i, IR-induced γ -H2AX is decreased by SIRT7 RNAi). By contrast, the effects on H2K122 succinylation occur only in response to DNA damage. This suggests that the effects on DNA damage response might not be directly due to the desuccinylation by SIRT7. This seems problematic for the author's model.

2. Expression of either the H3K122E succinyl mimic or the H3K122R non-succinylated mutant reduces NHEJ, HR, and DNA damage sensitivity (Fig. 5C, 6D). But in the MN assays (Figure 5A), H3K122E increases chromatin accessibility whereas H3K122R decreases accessibility. This suggests that the level of chromatin compaction doesn't correlate with the effects on repair/DNA damage sensitivity. How do the authors account for this?

3. A previous proteomic study for SIRT7 interacting proteins (Tsai et al, Mol. Cell Proteomics,

2012) identified a very different set of proteins (chromatin remodeling, ribosome biogenesis), versus the DNA repair proteins reported in this study. How do the authors account for this discrepancy? Did they detect any of the proteins from the previous study? Given the frequency with which the Kus and Parp are found in Flag-co-IPs, there is concern the interactions are not physiologic. The data should be confirmed by co-IP of the endogenous proteins.

4. The validation of the H3K122-Succ antibody by peptide competition does not rule out that the antibody might detect the peptide independent of succinylation. The authors should show competition (or lack of it) by the non-succinylated peptide.

5. To really prove that SIRT7 directly desuccinylates H3K122-succ in vitro, the authors should show some mass spec analysis in addition to westerns.

6. Did the authors test whether SIRT7 deacetylates H3K122Ac?

7. The introductory statement (and other comments in the introduction): "the intrinsic enzymatic activity of this sirtuin protein remains to be investigated and the cellular function of SIRT7 remains to be explored." incorrectly implies that nothing is known about SIRT7 enzymatic activity or cellular function.

Additional issues:

1. In the quantification of succinylation in Figure 1C, the error bars are very large and it is difficult to get a sense of the fold change in succinylation. Can the data be plotted on a more meaningful scale, and with comparisons to sites that were not significantly altered and negative controls.

2. The authors do not indicate why the SIRT7 S111A mutation (and the double S111A/H187A mutation) was studied, and they should provide a brief explanation in the text for what these mutations were intended to test. Mutation of the corresponding residue in other sirtuins was shown to reduce deacetylase activity (Min et al Cell 2001), and previous studies showed SIRT7 S111A impairs rDNA transcriptional activation by SIRT7 (Ford et al, Genes Dev 2006). The authors should discuss their observations that the mutation appears not to affect H2K122 desuccinulation or H3K18 deacetylation.

3. The deacetylase activity of SIRT7 on H3K18 has been shown to be more efficient on

nucleosomal histones than free histones from calf thymus (Gil et al, NAR 2013). It is difficult to compare the data for de-succinylation of H3K122 in these contexts in Figure 2 quantitatively (more SIRT7 was used in the nucleosome assays), but it seems that activity may be greater on free histones. A more quantitative comparison of the activities on nucleosomes versus CTHs should be straightforward, and could have implications for the contexts in which the de-succinylation actually occurs.

4. It would be nice to see if SIRT7 co-localizes with γ -H2AX at global DNA breaks induced by IR or other DSB agents?

5. Demonstration of Parp-1 dependence of SIRT7 relocalization to DSBs would be more convincing with specific Parp-1 inactivation by RNAi or knockout.

6. The authors should discuss how they envision chromatin compaction in response to desuccinylation might promote DNA repair? This is opposite of the model that decreased compaction can create accessibility for repair factors.

Reviewer #3 (Remarks to the Author):

In this manuscript, the authors reported sirt7 as a desuccinylase using in vitro enzymatic assay and validate this activity in cell lines by over expression and knocking down experiments. The authors further demonstrated that sirt7 was associated with DNA repair proteins by IP/MS experiments. Interestingly, sirt7 was recruited to DNA double strand breaks (DSB) by PARP1 and successful repair of DSB was dependent on desuccinylation of H3K122 by sirt7.

This is a nice story with significant novelty. The authors discovered that sirt7, one of the least studied sirtuin family members, had new activity- desuccinylase activity. They also provided functional studies and suggested that histone succinylation played an important role in DNA repair activity. Generally, the authors had proven their points by a series of in vitro and in cell line experiments. However, I suggest the authors to improve some functional studies before publication.

Major points:

1. Desuccinylase activity of sirt7:

In the in vitro enzymatic assay, sirt7 was shown to have desuccinylase activity toward H3K122succ peptide while sirt5 does not. Considering the well-studied desuccinylase activity of sirt5, this should be clarified. Is it possible that sirt5 and sirt7 having specificities toward

different succinylation sites? Is H3K122 the only site that sirt7 has activity in vitro? The authors are suggested to study more substrates to strengthen their point.

2. H3K122suc in DSB repair:

Sirt7 was shown to be recruited to DSB sites in a Parp1 dependent way. Is H3K122suc level decreased at the DSB sites where sirt7 was recruited? If yes, is the decrease of H3K122suc Parp1 dependent? The authors only studied H3K122suc level by western blot, which may not be specific to the DSB sites. Laser microirradiation and immunofluorescent assay using anti-H3K122suc or pan anti-Ksuc antibodies should be considered.

3. H3K122suc vs H3K122ac:

The authors showed that desuccinylase activity of sirt7 was indispensable for DSB repair. Does sirt7 have other activities such as deacetylase in addition to desuccinylase activity? Care should be taken that mutation of H3K122 to E/R has influence not only on succinylation but also on acetylation. The authors should study whether H3K122ac level is changed during DNA damage insult and whether sirt7 has deacetylase activity in vitro and in cell lines.

Minor points:

1. In Figure 1E, the authors are suggested to study immunofluorescence by over expression (OE) of Sirt7, showing sirt7 and lysine succinylation levels in the same cell, in a higher resolution picture.
2. In Figure 1D, the first panel indicates an uneven exposure for pan-succ signals, especially between sirt7wt and sirt7s111a.
3. In Figure 1F, the middle panel, the blot signals are too weak to demonstrate the antibody specificity clearly.
4. On page 12 of manuscript, the authors described "treatment with TSA or sodium butyrate had no effect on the level of H3K122succ, although treatment with NAM, TSA, or sodium butyrate all resulted in a marked increase in the level of H3 pan-acetylation (Figure 1H)". However, in Figure 1H, H3K122suc level was increased by either HDAC inhibitor treatment.
5. In Figure 4C, there is no labeling about which one is "NHEJ" or "HR".
6. "HBK20succ" is misspelled in Figure 4G.
7. In Introduction, wrong citation (#34) for sirt5 as an enzyme for malonylation.
8. In 1st paragraph of Result section page 8. Wrong citation . first report on Sirt 5 as a decylation enzyme appeared in Mol Cell Proteomics, 2011, 10, M111.012658.

Specific comments about statistics:

The statistical tests, the description of error bars and probability values are appropriate.

Summary of new data in the revised manuscript

1. According to the comment of reviewer #1, we further examined H3K122succ levels in SIRT7-depleted HCT116 and HeLa cells (Supplementary Figure 4b), as well as in SIRT7 knockout (KO) U2OS cells (Supplementary Figure 4c). In addition, considering that MTS assays were performed at 72 h after VP16 or IR treatment, a time might be too short to show the effect of IR and VP16 on cell survival, we thus used clonogenic survival assays to determine the effect of IR on cells survival in SIRT7-depleted HCT116, HepG2 cells, or SIRT7 KO U2OS cells (Figure 6d, 6e, and 6g). Therefore Figure 6b-d in the initial manuscript, which demonstrated the effect of SIRT7 on cell survival by MTS assay, has been moved to the file of “Figure for reviewer”.
2. According to the comment of reviewer #1, we have repeated the experiments described in Fig. 1D, and replaced the blots and added quantifications (Figure 1d and Supplementary Figure 2a).
3. According to the comment of reviewer #1, we have now added quantification for all westerns analysis of histone modification by densitometry (Figure 1d and Supplementary Figure 2a; Figure 1g and Supplementary Figure 4a; Supplementary Figure 4b; Supplementary Figure 4c; Figure 1h and Supplementary Figure 4d; Figure 2c-2d and Supplementary Figure 5a-5c; Figure 4h-i and Supplementary Figure 10a-10c; Supplementary Figure 13c).
4. Supplementary Figure 1 contains new data verifying the specificity of pan anti-succinyllysine rabbit antibodies.
4. Supplementary Figure 2b contains new data of immunofluorescent assays in GFP-SIRT7-overexpressing U2OS cells with pan-lysine succinylation polyclonal antibodies.

5. Supplementary Figure 3 contains new data verifying the specificity of anti-H3K122succ and anti-H2BK120succ antibodies.
6. Supplementary Figure 6 contains new data of mass spec analysis for SIRT7 *in vitro* activity assay.
7. Supplementary Figure 7 contains new data showing the effect of PARP1 knockdown on SIRT7 accumulation at laser-induced damage sites.
8. According to the comment of reviewer #1, the original FACs plots for both HR and NHEJ DNA repair upon SIRT7 knockdown in Fig. 4a and Fig. 4b have been provided in Supplementary Figure 8.
9. Supplementary Figure 9 contains the new data of dynamic recruitment of SIRT7 to DSB sites at different time points after I-SceI transfection by using ChIP assay.
10. Supplementary Figure 11 contains the new data demonstrating whether the decrease of H3K122succ is SIRT7 and PARP1-dependent and specific to the DSB sites by laser microirradiation and immunofluorescent assays.
11. Supplementary Figure 12a-b contains the new data supporting H3K122 associated with chromatin condensation by dose-, time-dependent MNase sensitivity assays and nucleosome stability assays.
12. Supplementary Figure 12c-d contains the new data supporting SIRT7 linked to chromatin compaction by the experiment of nuclear area comparison, and detailed quantification of proliferation patterns.
13. Supplementary Figure 13 contains the new data supporting Sirt7 associated with chromatin condensation by dose-, time-dependent MNase sensitivity assays and nucleosome stability assays.

RE: NCOMMS-16-00869

Title: SIRT7 Is a Histone Desuccinylase that Functionally Links to Chromatin Compaction and Genome Stability

Response to reviewers' comments-

Reviewer #1:

Major comments:

1. The rationale to use only MCF7 and 293T for the biochemical assays is not clear. Other lines exhibit equally “low” or “high” levels, and since the authors do not validate their experiments in more physiological conditions (in vivo), their results will be more convincing if replicated in other lines. In this context, the study will benefit if some of the claims made in knockdown cells will be shown in SIRT7 KO cells.

Authors: To comply with the reviewer's suggestions, we examined H3K122succ levels in control and SIRT7-depleted HCT116 and HeLa cells, and the results showed that knockdown of SIRT7 resulted in increase in H3K122succ levels in HCT116 and HeLa cells, with no obvious changes in H3K18ac, H3K122ac, and H2BK120succ levels (Supplementary Fig. 4b), supporting our observations in MCF-7 cells. Similar results were obtained in SIRT7 knockout (KO) U2OS cells (Supplementary Fig. 4c). In addition, clonogenic survival assays were also performed in control or SIRT7-depleted HCT116, HepG2 cells, or SIRT7 KO U2OS cells in

response to IR. The results showed that either SIRT7 knockdown or KO in above cell lines significantly compromised cell survival upon IR treatment (Fig. 6d, 6e, and 6g), consistent with the observations that we described in previous version of the manuscript. Moreover, the recruitment of SIRT7 at DSB sites (Fig. 3) and the MNase sensitivity assays (Fig. 5) were performed in U2OS cells, and the results are in agreement with our arguments.

2. Although the mass spectrometry assays seem solid, the biochemistry to support these results is not convincing. For instance, most of the westerns presented exhibit modest changes in the putative succinylation sites. It is well established that histone modifications are difficult to quantify in westerns, given intrinsic variability in signal. All these westerns should be performed at least in triplicates, and densitometry should be performed in order to provide a quantification of these experiments. As it stands, the data does not look convincing.

Particularly, Fig. 1D, it seems the line for the Pan-succ antibody has a bubble exactly for the Sirt7wt and SIRT7S11A extracts. Similarly, their newly generated antibodies should be further characterized. In Fig. 1F, the authors show that the presence of succ. peptides in competition assays completely eliminated the signal in the westerns, yet they did not show other peptides (with different modification, or even H3 unmodified peptides) as negative controls. The changes claimed in Fig. 1G are modest. Without quantification and replicates, these results are inconclusive. The same could be said for all the blots in Fig. 2C-D, and those in Fig. 4H-I.

Authors: All of the westerns for histone modifications were performed at least in triplicates. To address the reviewer's concerns, 1) We have now added quantification by densitometry (Fig. 1d and Supplementary Fig. 2a; Fig. 1g and Supplementary Fig. 4a; Supplementary Fig. 4b;

Supplementary Fig. 4c; Fig. 1h and Supplementary Fig. 4d; Fig. 2c-2d and Supplementary Fig. 5a-5c; Fig. 4h-i and Supplementary Fig. 10a-10c); 2) We have repeated the experiments described in Fig. 1D, and replaced the blots and added quantifications (Fig. 1d and Supplementary Fig. 2a); 3) The specificity of anti-H3K122succ and anti-H2BK120succ antibodies were characterized by dot blotting and western blotting competed with succinyl peptides, unmodified control peptides, or the malonyl peptides. The results showed that only H3K122succ peptides blocked the binding of anti-H3K122succ and only the H2BK120succ peptides blocked the binding of anti-H2BK120succ (Supplementary Fig. 3); 4) Although H3K122succ level had a modest 1.8 fold increase upon SIRT7 knockdown in MCF-7 cells in Fig. 1g, there was a more evident decrease of H3K122succ level upon overexpression of SIRT7 in HEK293T cells (Fig. 1g and Supplementary Fig. 4a). Similar results were obtained in HCT116 and HeLa cells upon SIRT7 knockdown (Supplementary Fig. 4b) and in SIRT7 KO U2OS cells (Supplementary Fig. 4c).

3. In this context, the western blot in Fig. 1H is worrisome. The authors claim that only NAM treatment increased H3K122succ levels, yet it seems that also TSA and Sodium butyrate increased its levels, a result that goes against their hypothesis. Of concern, treatment with TSA should majorly increase pan-acetylation, yet the western showed only a modest effect (despite the authors claiming "marked increase"). Such results raise concerns on the solidity of the whole western blot.

Authors: To address the reviewer's concerns, we have repeated the experiments in Fig. 1h. The results showed that while H3K122succ levels increased slightly under TSA and sodium

butyrate treatment, treatment with NAM was associated with a much more evident increase in H3K122succ level (Fig. 1h and Supplementary Fig. 4d). We have optimized the experiments concerning the effect of TSA treatment on pan-acetylation and observed a robust increase in pan-acetylation upon TSA treatment (Supplementary Fig. 4d). The effect of TSA and Sodium butyrate on H3K122succ levels might be indirect, as it is documented that various histone modifications interplay and are interdependent¹⁻³.

4. The IF shown in Fig. 1E is not convincing. Has this antibody been tested in IF assays before. The authors should include controls to verify the antibody truly recognize histone succinylation.

Authors: The antibodies are commercially available and have been tested for IF before. To comply with the reviewer's requests, we have added to the revision the experiments verifying the specificity of pan anti-succinyllysine rabbit antibodies including dot blot assays with serial dilution of various modified peptide libraries, unmodified peptide library and 9 site succinylated peptides probed with anti-succinyllysine rabbit antibodies alone or pre-adsorbed with various modified peptide libraries and unmodified peptide library (Supplementary Fig. 1a-e). We have also repeated IF and replaced the figures (Fig. 1e) in the revised manuscript.

5. Loading controls and quantification should be included in the dot blot assays in Fig. 2B.

Authors: Loading controls and quantification have been added to Fig. 2b.

6. The data in Fig. 3, indicating recruitment of SIRT7 to sites of DSBs is compelling.

Authors: We appreciate the reviewer for this comment.

7. The experiments shown in Fig.4 demonstrating a role for SIRT7 in both HR and NEHJ DNA repair seem robust. It will be reassuring to see the original FACs plots for at least some of these experiments.

Authors: Original FACs plots for both HR and NHEJ DNA repair upon SIRT7 knockdown in Fig. 4a and Fig. 4b have been provided in Supplementary Fig. 8a-8b.

8. However, in this context, how long after ISceI transfection was the experiment in figure 4J done? It is very difficult to assess any kinetics in this setting since this system is based on cut-repair cycles in the presence of the ISceI enzyme.

Authors: In DR-GFP-U2OS cells used in our cell flow cytometry and ChIP assays, the reporter, DR-GFP, contains a GFP-encoding cDNA that has an endogenous BcgI restriction site replaced by an I-SceI restriction site, thereby rendering it non-functional. A DSB is induced by transfection of cells with a plasmid that encodes I-SceI enzyme to cut the I-SceI site. An incomplete GFP sequence is located downstream and can serve as a donor for intra-chromosomal homologous recombination. Repair by HR restores a functional GFP cDNA with the original BcgI site. So after HR repair, the I-SceI restriction site was replaced by BcgI restriction site⁴⁻⁶. Therefore, there're no cut-repair cycles in this DR-GFP systems. Considering that the recruitment of SIRT7 to DSB sites is transient, we tested the recruitment of SIRT7 to

DSB sites at different time points after I-SceI transfection by using ChIP assay before the experiments described in Fig.4d and 4j. Briefly, DR-GFP-U2OS cells transfected with I-SceI were collected at different time points and subjected to qChIP analysis with antibodies against SIRT7. The results showed that SIRT7 had maximal recruitment to DSB sites at 40 h after transfection in our experiment (Supplementary Fig. 9). So we chose this time point for subsequent qChIP experiments in Fig.4d and 4j.

9. Although the recruitment of SIRT7 to sites of breaks is clear, the mechanistic roles for such recruitment are less clear. In this context, the experiments provided to claim a role in modulating chromatin compaction are much less convincing. MNase experiments are known to have high variability and, most of the times, is very difficult to get informative results. To overcome this, it is imperative to repeat the experiment several times using different amounts of MNase and times of digestion to see a pattern of more digested chromatin with increasing units of enzyme or time of digestion (as a control that the experiment is working). It is surprising then that the authors conclude that H3K122succ and Sirt7 regulate chromatin condensation by doing just one experiment, in which this reviewer cannot see any significant difference. Moreover, if H3K122succ induces chromatin relaxation and this mark decreases after 1h of IR (fig 4I), why is chromatin in a more open state (fig 5b)?

Authors: To comply with the reviewer's requests, we have designed and performed a serial of additional experiments. 1) Using different amount of MNase (20, 60, 100 gel unit) in MNase experiment, we found that chromatin with H3K122E mutant (H3K122succ mimic) is more sensitive to MNase digestion (Supplementary Fig. 12a), suggesting that H3K122E mutation

resulted in a loose chromatin structure; 2) MNase experiments in control or SIRT7-depleted U2OS cells using different amount of MNase upon IR treatment showed that SIRT7 depletion significantly increased MNase sensitivity of chromatin (Supplementary Fig. 13a-b); 3) To further confirm the effect of H3K122succ on chromatin condensation, nucleosome stability assay was performed to analyze the difference of histone-DNA interactions between H3wt-, H3K122E-, and H3K122R-stably expressing U2OS cells. The results showed that H3K122E (H3K122succ mimic) was associated with a reduced histone-DNA interaction while H3K122R (succinylation resistant mutant) was associated with a stabilized nucleosomes (Supplementary Fig. 12b), consistent with our hypothesis that H3K122 succinylation is associated with an increase in chromatin accessibility and desuccinylation of H3K122succ is linked to a condensed chromatin state; 4) Overexpression of GFP-SIRT7 resulted in smaller nuclear size compared to cells expressing GFP only (Supplementary Fig. 12c); 5) Detailed quantification of proliferation patterns in shControl and shSIRT7 cells demonstrated that, although cells with SIRT7 knockdown had a similar number of cells in S phase compare to control cells, they have a lower population of cells in later S phase than control cells (Supplementary Fig. 12d); 6) Nucleosome stability assays showed that NaCl solubility increased for γ H2AX, H2AX, and H3 upon DNA damage, and SIRT7 depletion further enhanced NaCl solubility of histones at 0.25, 1, 4, and 8 h after IR and delayed their recovery (Supplementary Fig. 13c), consistent with the results of MNase assay in Fig. 5b. These observations altogether point to a role of SIRT7-mediated H3K122succ desuccinylation in chromatin compaction.

Regarding to the question in Fig. 4i and Fig. 5b, it has been reported that chromatin is hypersensitive to nuclease digestion following exposure to IR⁷⁻⁹. The extent of the increased sensitivity to nuclease digestion indicates that the decrease in nucleosome compaction in

response to DSBs affects a significant fraction of the chromatin¹⁰. This is consistent with our observations from MNase experiments (Fig. 5b, and Supplementary Fig. 13a) that SIRT7 knockdown (KD) resulted in chromatin hypersensitivity to MNase digestion upon IR treatment when compared to controls. The temporal coordination of chromatin dynamics might be complemented by spatial separation of relaxed and condensed chromatin domains in the vicinity of a DSB, with relaxed and compact regions playing distinct roles in DNA damage response (DDR) signaling and repair¹¹. In fact, subcompartments within DNA damage foci have been observed by superresolution microscopy and their chromatin environments proposed to be distinct¹². Western blotting analysis of H3K122succ levels and MNase sensitivity assays might only reflect the different sides of the cells in response to IR; the global decrease of H3K122succ could be accompanied by local increase of H3K122succ in chromatin, and global increase of MN sensitivity could be also accompanied by local decrease of MN sensitivity. Moreover, since there are many other histone modifications associated with chromatin compaction and decompaction involved in DDR¹³⁻¹⁵, the overall chromatin structure could not be decided by some or other histone modifications.

10. Assuming that SIRT7-dependent desuccinylation indeed enhances chromatin compaction, how such compaction will benefit DNA repair? One would think that increased compaction may decrease recruitment of repair factors, in turn negatively influencing DNA repair, which is the opposite to what the authors claim.

Authors: It has been reported that chromatin condensation is a transient but integral part of DNA damage response, and that condensed chromatin enhances upstream signaling, thus

promotes DNA repair¹¹, whereas persistent condensation inhibits downstream repair and recovery¹¹, which is consistent with our results in Fig. 5c that both persistent succinylation (H3K122E) and persistent non-succinylation (H3K122R) impaired HR and NHEJ repair efficiency. In addition, it has been proposed that a repressive chromatin domain is required for inhibiting local transcription, compacting local chromatin structure, and rewriting the local epigenetic landscape during the priming state of DNA repair¹⁶. Thus, linking SIRT7-mediated transient H3K122succ desuccinylation at DSB site to early stage of DDR is consistent with these studies.

11. In Fig. 5C, why do the control “H3wt” cells exhibit 15% GFP cells, while in the previous Fig. 4A, control cells showed around 8-10% GFP? Such discrepancy in control assays raise concerns about reproducibility, in particular when cells expressing the transfecting mutant histones exhibit ~8-10% GFP, which are levels of control cells in the previous experiment.

Authors: The cells in Fig. 4a were EJ5-HEK293 cells transfected with siRNAs and the cells in Fig. 5c were EJ5-HEK293 cells stably expressing H3wt, H3K122E, and H3K122R. The differences mentioned by the reviewer could be due to technical variations including the variation of transfection efficiency, etc. Although the GFP percentages varies in different experiments, the pattern of the GFP percentage in H3wt, H3K122E, and H3K122R expressing cells was similar: decreases in NHEJ repair efficiency were detected in EJ5-HEK293 cells expressing H3K122E or H3K122R compare to cells expressing H3wt in triplicate experiments (Fig. 5c and Supplementary Fig. 14a-b).

12. In Figure 6 B-D: the DNA damage sensitivity assays show a very modest effect (please note the scale is set up to enhance the difference. Survival assays should be plotted in log scale). P-values should be indicated to determine whether the difference claimed is significant. Can these experiments be repeated in other lines?

Authors: Survival curves from MTS assays have been plotted in log scale and p-values have been added (Figures to reviewer). Considering that MTS assays were performed at 72 h after VP16 or IR treatment, a time might be too short to show the effect of IR and VP16 on cell survival, we thus used clonogenic survival assays to determine the effect of IR on cells survival as reported in other studies¹⁶⁻¹⁹, in which control or SIRT7-depleted MCF-7, U2OS cells and U2OS cells stably expressing H3wt, H3K122E or H3K122R were treated with or without IR at the indicated doses and then subjected to clonogenic survival assays after 10 days (Fig. 6b-c, Fig. 6f). Similar experiments were also repeated in HCT116, HepG2 and SIRT7 knockout U2OS cells (Fig. 6d-e, Fig. 6g).

Minor concerns:

1. Few grammatical errors should be corrected. For instance, on page 4 “in last decade” should be “in the last decade”. On page 20, “a residue could not be succinylated” should be “a residue that could not be succinylated”.

Authors: We appreciate the reviewer and have corrected the writings.

RE: NCOMMS-16-00869

Title: SIRT7 Is a Histone Desuccinylase that Functionally Links to Chromatin Compaction and Genome Stability

Response to reviewers' comments-

Reviewer #2:

Major concerns:

1. In Figure 4H and I, SIRT7 RNAi seems to increase baseline γ -H2AX but does not increase γ -H2AX upon DNA damage (in fact, in 4i, IR-induced γ -H2AX is decreased by SIRT7 RNAi). By contrast, the effects on H2K122 succinylation occur only in response to DNA damage. This suggests that the effects on DNA damage response might not be directly due to the desuccinylation by SIRT7. This seems problematic for the author's model.

Authors: γ H2AX was originally identified as an early event after the direct formation of DNA double-strand breaks (DSBs) by ionizing radiation²⁰ and now the generation of γ H2AX is also believed to occur in association with secondarily formed DSBs by endogenous genotoxin or cellular processing such as reactive oxygen species (ROS) produced during normal cell metabolism, DNA replication and repair at the site of the initial damage, including DNA adducts, crosslinks, and UV-induced photolesions²¹. It is estimated that in normal human cells approximately 1% of single-strand lesions are converted to approximately 50 endogenous DSBs per cell per cell cycle²². Thus, SIRT7 might play a role in response to these damages, a

possible reason why SIRT7 RNAi seems to increase baseline γ H2AX (Fig. 4h-i and Supplementary Fig. 10a-c). Since the occurrence of endogenous DSB is much lower than that of the DSBs induced by the DNA damage reagents or IR treatment, the changes of desuccinylation of H3K122succ catalyzed by SIRT7 might be too weak to be detected during endogenous DSB repair, a possible explanation for the deviation between the changes of H3K122 succinylation and γ H2AX during endogenous DSB repair. In responding to the reviewer's concerns, all of the experiments in Fig. 4h and Fig. 4i have been repeated at least in triplicate, and the results showed that SIRT7 knockdown was associated with γ H2AX level increment in response to CPT and IR treatment (Fig. 4h and Supplementary Fig. 10a-c).

2. Expression of either the H3K122E succinyl mimic or the H3K122R non-succinylated mutant reduces NHEJ, HR, and DNA damage sensitivity (Fig. 5C, 6D). But in the MN assays (Figure 5A), H3K122E increases chromatin accessibility whereas H3K122R decreases accessibility. This suggests that the level of chromatin compaction doesn't correlate with the effects on repair/DNA damage sensitivity. How do the authors account for this?

Authors: Our data showed the recruitment of SIRT7 to DNA damage sites is a transient process, and that SIRT7 mediated transient H3K122succ desuccinylation at DSB site in the early stage of DDR promotes DNA repair by promoting chromatin compaction. Though little is known about the mechanisms of chromatin condensation during DNA repair, it is believed that chromatin condensation is a transient but integral part of the DNA damage response; condensed chromatin enhances upstream signaling, whereas persistent condensation inhibits downstream repair and recovery¹¹. In Fig. 5c and 6f, stably expressed H3K122E and H3K122R

mimic persistent succinylation and persistent non-succinylation of H3K122, respectively, but they could not mimic the dynamics of SIRT7-mediated transient H3K122succ desuccinylation. Thus, the expression of H3K122E succinyl mimic as well as H3K122R succinylation-resistant mutant reduces NHEJ, HR, and increases DNA damage sensitivity (Fig. 5c and Fig. 6f), by contrast to the results of the MNsae assays (Fig. 5a and Supplementary Fig. 12a) in which H3K122E increases chromatin accessibility whereas H3K122R decreases its accessibility.

3. A previous proteomic study for SIRT7 interacting proteins (Tsai et al, Mol. Cell Proteomics, 2012) identified a very different set of proteins (chromatin remodeling, ribosome biogenesis), versus the DNA repair proteins reported in this study. How do the authors account for this discrepancy? Did they detect any of the proteins from the previous study? Given the frequency with which the Kus and Parp are found in Flag-co-IPs, there is concern the interactions are not physiologic. The data should be confirmed by co-IP of the endogenous proteins.

Authors: We did identify SIRT7 interacting proteins functioning in chromatin remodeling and ribosome biogenesis beside the DNA repair proteins, including CHD4, MTA2, RBBP7, RBBP4, MTA2, and CHD7 etc. involving in chromatin remodeling, and GNL2, GTPBP4, URB2, Nucleolar pre-ribosomal-associated protein 1, 60S ribosomal protein L5, 40S ribosomal protein S6 etc. related with ribosome functioning. These proteins are shown in Fig. 3a. The endogenous Co-IP between PARP1 and SIRT7 is presented in Fig. 3d.

4. The validation of the H3K122-Succ antibody by peptide competition does not rule out that the antibody might detect the peptide independent of succinylation. The authors should show competition (or lack of it) by the non-succinylated peptide.

Authors: The specificity of anti-H3K122succ and anti-H2BK120succ antibodies were characterized by dot blotting and western blotting competed with succinyl peptides, unmodified control peptides, or malonyl peptides. The results showed that only the H3K122succ peptide blocked the binding of anti-H3K122succ and only the H2BK120succ peptide blocked the binding of anti-H2BK120succ (Supplementary Fig. 3).

5. To really prove that SIRT7 directly desuccinylates H3K122-succ in vitro, the authors should show some mass spec analysis in addition to westerns.

Authors: *in vitro* assays with calf thymus histones and purified SIRT7wt in the presence of NAD⁺ were analyzed by mass spec. The results show that SIRT7 had strong H3K122succ desuccinylase activity, whereas the level of H2BK120succ, H3K122ac, and H3K18ac had no detectable change (Supplementary Fig. 6a-6c).

6. Did the authors test whether SIRT7 deacetylates H3K122Ac?

Authors: The deacetylation activity for H3K122ac by SIRT7 was determined by following methods: 1) SIRT7 was overexpressed in HEK293T cells or knocked down in MCF-7, HCT116, HeLa cells or knockout in U2OS cells. Histones were extracted and H3K122ac were

analyzed by western blotting. The results showed that the levels of H3K122ac were unaffected in these cells, regardless of overexpression or knockdown of SIRT7 (Fig. 1g and Supplementary Fig. 4a-4c); 2) *in vitro* desuccinylation assays using SIRT7wt/H187Y and calf thymus histones or mononucleosomes were analyzed for the H3K122ac levels by western blotting and mass spec. The results indicated that SIRT7wt showed no H3K122ac deacetylation activity toward calf thymus histones and mononucleosomes (Fig. 2c-d and Supplementary Fig. 5 and 6).

7. The introductory statement (and other comments in the introduction): "the intrinsic enzymatic activity of this sirtuin protein remains to be investigated and the cellular function of SIRT7 remains to be explored." incorrectly implies that nothing is known about SIRT7 enzymatic activity or cellular function.

Authors: We have modified the relevant statements.

Additional issues:

1. In the quantification of succinylation in Figure 1C, the error bars are very large and it is difficult to get a sense of the fold change in succinylation. Can the data be plotted on a more meaningful scale, and with comparisons to sites that were not significantly altered and negative controls.

Authors: According to the reviewer's suggestion, the mass spec data have been plotted on the fold change of control/knockdown of the experiments in triplicate (Fig.1c), with comparisons to sites that were not significantly altered.

2. The authors do not indicate why the SIRT7 S111A mutation (and the double S111A/H187A mutation) was studied, and they should provide a brief explanation in the text for what these mutations were intended to test. Mutation of the corresponding residue in other sirtuins was shown to reduce deacetylase activity (Min et al Cell 2001), and previous studies showed SIRT7 S111A impairs rDNA transcriptional activation by SIRT7 (Ford et al, Genes Dev 2006). The authors should discuss their observations that the mutation appears not to affect H2K122 desuccinulation or H3K18 deacetylation.

Authors: We appreciate the reviewer's suggestion and have added to the text the explanations for what these mutations were intended to. We have also added discussion of our observations related to the published results.

3. The deacetylase activity of SIRT7 on H3K18 has been shown to be more efficient on nucleosomal histones than free histones from calf thymus (Gil et al, NAR 2013). It is difficult compare the data for de-succinylation of H3K122 in these contexts in Figure 2 quantitatively (more SIRT7 was used in the nucleosome assays), but it seems that activity may be greater on free histones. A more quantitative comparison of the activities on nucleosomes versus CTHs should be straightforward, and could have implications for the contexts in which the de-succinylation actually occurs.

Authors: To comply with the reviewer's requests, *in vitro* desuccinylation assays were carried out with different amounts of purified FLAG-SIRT7wt and 1 μg of calf thymus histones or 1 μg of HeLa cells-derived mononucleosomes in the presence or absence of 1.0 mM NAD^+ and/or 10 mM NAM. The reaction mixture was analyzed by western blotting and the densities were quantified by Image J. The results showed no obvious substrates preference for SIRT7 between nucleosomes and CTHs (Supplementary Fig. 5c). Though the deacetylase activity of SIRT7 on H3K18ac has been shown to be more efficient on nucleosomal histones than free histones from calf thymus²³, succinyl group is different from acetyl group, and H3K122 is located on the lateral surface of the histone globular domain closing to the dyad symmetry axis. The DNA wrapped around the globular domain might affect the substrate affinity of SIRT7, and hence its desuccinylase activity when mononucleosomes were used as substrates.

4. It would be nice to see if SIRT7 co-localizes with γ -H2AX at global DNA breaks induced by IR or other DSB agents?

Authors: We agree with the reviewer. However, despite extensive efforts, the antibodies against SIRT7 from almost every company did not work for foci stain, although the antibodies worked for stain SIRT7 in nucleolus and laser path. U2OS cells transfected with GFP-SIRT7 were also used to perform this experiment, but the cells with GFP-SIRT7 overexpression had too high background in nucleus to detect the foci.

5. Demonstration of Parp-1 dependence of SIRT7 relocalization to DSBs would be more convincing with specific Parp-1 inactivation by RNAi or knockout.

Authors: In responding to the reviewer's suggestion, laser microirradiation and immunofluorescent assays have been performed upon PARP1 knockdown in U2OS cells. The results showed that, compare to siControl cells, siPARP1 resulted in a complete abrogation of SIRT7 accumulation at laser-induced damage sites 5 min after microirradiation (Supplementary Fig. 7a-7b).

6. The authors should discuss how they envision chromatin compaction in response to desuccinylation might promote DNA repair? This is opposite of the model that decreased compaction can create accessibility for repair factors.

Authors: As we mentioned early, although little is known about the mechanisms of chromatin condensation during DNA repair process, it has been shown that chromatin condensation is a transient but integral part of the DNA damage response, and that condensed chromatin enhances upstream signaling, thus promote DNA repair, whereas persistent condensation inhibits downstream repair and recovery¹¹, which is consistent with our results in Fig. 5c that both persistent succinylation (H3K122E) and persistent non-succinylation (H3K122R) impaired HR and NHEJ repair efficiency. In addition, it is proposed that that the requirement of repressive chromatin might be related with its' roles in inhibiting local transcription, compacting the local chromatin structure, and rewriting the local epigenetic landscape, stabilizing open chromatin structures and limiting DSB mobility during the initial moments

following DSB production¹⁶. The information has been added to discussion.

RE: NCOMMS-16-00869

Title: SIRT7 Is a Histone Desuccinylase that Functionally Links to Chromatin Compaction and Genome Stability

Response to reviewers' comments-

Response to Reviewer #3:

Major points:

1. Desuccinylase activity of sirt7: In the in vitro enzymatic assay, sirt7 was shown to have desuccinylase activity toward H3K122succ peptide while sirt5 does not. Considering the well-studied desuccinylase activity of sirt5, this should be clarified. Is it possible that sirt5 and sirt7 having specificities toward different succinylation sites? Is H3K122 the only site that sirt7 has activity in vitro? The authors are suggested to study more substrates to strengthen their point.

Authors: *in vitro* desuccinylation assays with synthesized peptides demonstrated that both SIRT7 and SIRT5 exhibit desuccinylase activity toward H3K122succ peptide (Fig. 2b), whereas only SIRT7 showed desuccinylase activity toward calf thymus histones (Fig. 2c, Supplementary Fig. 5a) and mononucleosomes (Fig. 2d, Supplementary Fig. 5b). As discussed

on page 27, although SIRT5 has been reported to regulate succinylation, malonylation, and glutarylation of both intra- and extra-mitochondrial proteins including histones²⁴⁻²⁷ and hydrolysis of succinyl of H3K9 succinyl peptide in *in vitro* desuccinylation assay²⁵, its mitochondrial subcellular localization excludes the possibility for this sirtuin protein to act on nuclear histones under physiological conditions.

Based on the reviewer's suggestions, *in vitro* desuccinylation assays were carried out with calf thymus histones and purified SIRT7wt in the presence of NAD⁺. Mass spec analysis revealed that besides H3K122succ, SIRT7 also desuccinylates H3K56succ and H3K79succ (Supplementary Fig. 6). As there are no available antibodies against these two modifications, we could not verify these desuccinylase activities by western blotting at present.

2. H3K122suc in DSB repair: Sirt7 was shown to be recruited to DSB sites in a Parp1 dependent way. Is H3K122suc level decreased at the DSB sites where sirt7 was recruited? If yes, is the decrease of H3K122suc Parp1 dependent? The authors only studied H3K122suc level by western blot, which may not be specific to the DSB sites. Laser microirradiation and immunofluorescent assay using anti-H3K122suc or pan anti-Ksuc antibodies should be considered.

Authors: To comply with the reviewer's requests, laser microirradiation and immunofluorescent assays have been performed using anti-H3K122suc antibody. Briefly, we analyzed H3K122succ level at DSB sites in control and SIRT7-depleted U2OS cells. The results showed that, in control U2OS cells, H3K122succ level at DSB sites decreased 5 min

after microirradiation compared to non-microirradiated regions, while in SIRT7-depleted U2OS cells, H3K122succ level at DSB sites did not change (Supplementary Fig. 11a). Furthermore, we analyzed H3K122succ level at DSB sites in U2OS cells transfected with siControl or siPARP1. The results showed that, in siControl cells, H3K122succ level at DSB sites decreased 5 min after microirradiation compared to non-microirradiated regions, while in siPARP1 cells, H3K122succ level at DSB sites had no detectable change (Supplementary Fig. 11b). These results suggest that the decrease of H3K122succ level is SIRT7- and PARP1-dependent and specific to DSB sites.

3. H3K122suc vs H3K122ac: The authors showed that desuccinylase activity of sirt7 was indispensable for DSB repair. Does sirt7 have other activities such as deacetylase in addition to desuccinylase activity? Care should be taken that mutation of H3K122 to E/R has influence not only on succinylation but also on acetylation. The authors should study whether H3K122ac level is changed during DNA damage insult and whether sirt7 has deacetylase activity in vitro and in cell lines.

Authors: To address the reviewer's concerns, the level of H3K122ac was detected upon SIRT7 overexpression in HEK293T cells or knockdown in MCF-7, HCT116, HeLa cells or knockout in U2OS cells. The results showed that the levels of H3K122ac were unaffected in these cells, regardless of overexpression or knockdown, or even knockout of SIRT7 (Fig. 1g and Supplementary Fig. 4). In addition, *in vitro* desuccinylation assays were carried out using SIRT7wt/H187Y and calf thymus histones or mononucleosomes, and the H3K122ac levels were analyzed by western blotting and mass spec. The results indicated that SIRT7wt showed

no detectable H3K122ac deacetylation activity toward calf thymus histones and mononucleosomes (Fig. 2b-c and Supplementary Fig. 5 and 6). Moreover, H3K122ac levels were also determined in MCF -7 and U2OS cells treated with or without CPT, VP16, or IR, the results revealed that H3K122ac levels showed no changes upon genotoxic insults (Fig. 4g-i and Supplementary Fig. 10a-c).

Minor points:

1. In Figure 1E, the authors are suggested to study immunofluorescence by over expression (OE) of Sirt7, showing sirt7 and lysine succinylation levels in the same cell, in a higher resolution picture.

Authors: To follow the reviewer's suggestion, immunofluorescence assays were performed by overexpressing GFP-SIRT7 in U2OS cells and staining with pan-lysine succinylation polyclonal antibodies. The results showed that, compared to cells without overexpression, pan-lysine succinylation levels were decreased in U2OS cells with overexpression of GFP-SIRT7 (Supplementary Fig. 2b)

2. In Figure 1D, the first panel indicates an uneven exposure for pan-succ signals, especially between sirt7wt and sirt7s111a.

Authors: The experiment has been repeated and figure has been replaced (Fig. 1d and Supplementary Fig. 2a).

3. In Figure 1F, the middle panel, the blot signals are too weak to demonstrate the antibody specificity clearly.

Authors: The experiments in Fig. 1f have been repeated and the figures have been replaced in the revised manuscript. In addition, the specificity of anti-H3K122succ and anti-H2BK120succ antibodies were further characterized by dot blotting and western blotting competed with succinyl peptides, unmodified control peptides, or malonyl peptides. The results showed that only H3K122succ peptide blocked the binding of anti-H3K122succ and only H2BK120succ peptide blocked the binding of anti-H2BK120succ (Supplementary Fig. 3).

4. On page 12 of manuscript, the authors described "treatment with TSA or sodium butyrate had no effect on the level of H3K122succ, although treatment with NAM, TSA, or sodium butyrate all resulted in a marked increase in the level of H3 pan-acetylation (Figure 1H)". However, in Figure 1H, H3K122succ level was increased by either HDAC inhibitor treatment.

Authors: We found that, under certain conditions, H3K122succ levels alters, although modestly, under TSA or sodium butyrate treatment. We speculate that the effect of TSA and sodium butyrate on H3K122succ levels might be indirect, as it is well documented that various histone modifications interplay and are interdependent¹⁻³.

5. In Figure 4C, there is no labeling about which one is "NHEJ" or "HR".

Authors: Labels have been added.

6. "HBK20succ" is misspelled in Figure 4G.

Authors: The spelling has been corrected.

7. In Introduction, wrong citation (#34) for sirt5 as an enzyme for malonylation.

Authors: The citation has been corrected.

8. In 1st paragraph of Result section page 8. Wrong citation. first report on Sirt 5 as a decylation enzyme appeared in Mol Cell Proteomics, 2011, 10, M111.012658.

Authors: The citation has been corrected.

References

- 1 Latham, J. A. & Dent, S. Y. Cross-regulation of histone modifications. *Nature structural & molecular biology* **14**, 1017-1024 (2007).
- 2 Suganuma, T. & Workman, J. L. Crosstalk among Histone Modifications. *Cell* **135**, 604-607 (2008).
- 3 Rothbart, S. B. & Strahl, B. D. Interpreting the language of histone and DNA modifications. *Biochimica et biophysica acta* **1839**, 627-643 (2014).

- 4 Pierce, A. J., Johnson, R. D., Thompson, L. H. & Jasin, M. XRCC3 promotes homology-directed repair of DNA damage in mammalian cells. *Genes & development* **13**, 2633-2638 (1999).
- 5 Li, X. *et al.* Histone demethylase KDM5B is a key regulator of genome stability. *Proceedings of the National Academy of Sciences of the United States of America* **111**, 7096-7101 (2014).
- 6 Oberdoerffer, P. *et al.* SIRT1 redistribution on chromatin promotes genomic stability but alters gene expression during aging. *Cell* **135**, 907-918 (2008).
- 7 Rubbi, C. P. & Milner, J. p53 is a chromatin accessibility factor for nucleotide excision repair of DNA damage. *The EMBO journal* **22**, 975-986 (2003).
- 8 Ziv, Y. *et al.* Chromatin relaxation in response to DNA double-strand breaks is modulated by a novel ATM- and KAP-1 dependent pathway. *Nature cell biology* **8**, 870-876 (2006).
- 9 Telford, D. J. & Stewart, B. W. Micrococcal nuclease: its specificity and use for chromatin analysis. *The International journal of biochemistry* **21**, 127-137 (1989).
- 10 Xu, Y. & Price, B. D. Chromatin dynamics and the repair of DNA double strand breaks. *Cell cycle* **10**, 261-267 (2011).
- 11 Burgess, R. C., Burman, B., Kruhlak, M. J. & Misteli, T. Activation of DNA damage response signaling by condensed chromatin. *Cell reports* **9**, 1703-1717 (2014).
- 12 Chapman, J. R., Sossick, A. J., Boulton, S. J. & Jackson, S. P. BRCA1-associated exclusion of 53BP1 from DNA damage sites underlies temporal control of DNA repair. *Journal of cell science* **125**, 3529-3534 (2012).

- 13 Polo, S. E. & Almouzni, G. Chromatin dynamics after DNA damage: The legacy of the access-repair-restore model. *DNA repair* (2015).
- 14 Shi, L. & Oberdoerffer, P. Chromatin dynamics in DNA double-strand break repair. *Biochimica et biophysica acta* **1819**, 811-819 (2012).
- 15 Price, B. D. & D'Andrea, A. D. Chromatin remodeling at DNA double-strand breaks. *Cell* **152**, 1344-1354 (2013).
- 16 Ayrapetov, M. K., Gursoy-Yuzugullu, O., Xu, C., Xu, Y. & Price, B. D. DNA double-strand breaks promote methylation of histone H3 on lysine 9 and transient formation of repressive chromatin. *Proc Natl Acad Sci U S A* **111**, 9169-9174 (2014).
- 17 Li, D. Q. *et al.* MORC2 signaling integrates phosphorylation-dependent, ATPase-coupled chromatin remodeling during the DNA damage response. *Cell reports* **2**, 1657-1669 (2012).
- 18 Xu, Y. *et al.* Histone H2A.Z Controls a Critical Chromatin Remodeling Step Required for DNA Double-Strand Break Repair. *Molecular Cell* **48**, 723-733 (2012).
- 19 Ochi, T. *et al.* DNA repair. PAXX, a paralog of XRCC4 and XLF, interacts with Ku to promote DNA double-strand break repair. *Science* **347**, 185-188 (2015).
- 20 Rogakou, E. P., Pilch, D. R., Orr, A. H., Ivanova, V. S. & Bonner, W. M. DNA double-stranded breaks induce histone H2AX phosphorylation on serine 139. *The Journal of biological chemistry* **273**, 5858-5868 (1998).
- 21 Bonner, W. M. *et al.* GammaH2AX and cancer. *Nature reviews. Cancer* **8**, 957-967 (2008).

- 22 Vilenchik, M. M. & Knudson, A. G. Endogenous DNA double-strand breaks: production, fidelity of repair, and induction of cancer. *Proceedings of the National Academy of Sciences of the United States of America* **100**, 12871-12876 (2003).
- 23 Tong, Z. *et al.* SIRT7 Is Activated by DNA and Deacetylates Histone H3 in the Chromatin Context. *ACS chemical biology* **11**, 742-747 (2016).
- 24 Peng, C. *et al.* The first identification of lysine malonylation substrates and its regulatory enzyme. *Molecular & cellular proteomics : MCP* **10**, M111 012658 (2011).
- 25 Du, J. T. *et al.* Sirt5 Is a NAD-Dependent Protein Lysine Demalonylase and Desuccinylase. *Science* **334**, 806-809 (2011).
- 26 Rardin, M. J. *et al.* SIRT5 regulates the mitochondrial lysine succinylome and metabolic networks. *Cell metabolism* **18**, 920-933 (2013).
- 27 Park, J. *et al.* SIRT5-mediated lysine desuccinylation impacts diverse metabolic pathways. *Molecular cell* **50**, 919-930 (2013).

b

c

d

(b) MCF-7 cells with or without SIRT7 knockdown were treated with increasing concentrations of VP16 for 72 h and subjected to MTS assays for cell survival. Error bars represent the mean \pm SD of triplicate experiments. (c) Control or SIRT7-depleted U2OS were treated with IR and subjected to MTS assays for cell survival. Error bars represent the mean \pm SD of triplicate experiments. (d) U2OS cells stably expressing H3wt, H3K122E or H3K122R were treated with IR and subjected to MTS assays for cell survival. Error bars represent mean \pm SD of triplicate experiments. * $p < 0.05$ and ** $p < 0.01$ (two-tailed unpaired Student's t test).

REVIEWERS' COMMENTS:

Reviewer #1 (Remarks to the Author):

In this revised manuscript, Yu and colleagues did major efforts to respond to the reviewers concerns, and the manuscript is highly improved. In particular the new data in other cells, the higher quality and the quantification of the westerns, and the clonogenic survival assays look very convincing. One minor comment, for fairness, it is surprising the authors do not include in their Discussion the fact that other sirtuins have been previously shown to be recruited to DBSs (see the papers from Tsai and colleagues on Sirt1 and Mostoslavsky and colleagues for SIRT6). Adding those to the Discussion, and putting Sirt7 in that context, will be reassuring. The manuscript warrant publication, and it will be of broad interest to researchers in the field of genomic instability and sirtuins.

Reviewer #2 (Remarks to the Author):

Li et al, Nature Comm 2016

In this revised manuscript and response the authors have made good efforts to address the previous reviews, and they have adequately answered my concerns. The identification of SIRT7 as a histone desuccinylase is novel and of strong general interest. I have one final issue and some minor textual suggestions.

Major issue

As indicated by another reviewer, the authors did not consider contributions of other SIRT7 substrates to the DNA damage response biology. In the revision, they have now added data showing possible additional desuccinylation sites targeted by SIRT7, and this is interesting. I also agree with the authors that given the lack of antibodies to these sites, further characterization of these sites is not necessary for this manuscript, which is focused on H3K122succ. However, SIRT7 does have additional substrates, namely H3K18Ac, several non-histone acetylated substrates, and presumably additional ones yet-to-be identified. It is possible that some of the effects of SIRT7 on DNA damage responses and DNA repair might be mediated by such other substrates. While I believe analysis of such models is beyond the scope of this study, it is important that the authors include a brief discussion that other SIRT7 substrates might be involved in the DNA repair biology of SIRT7. This would not detract from the current findings, and in fact would add interest that SIRT7 may coordinate multiple aspects of DNA

damage biology.

Minor issues:

The following introductory statements in the are inaccurate and should be revised:

1. Line 99: "Silent information regulator 2 (Sir2) proteins, or sirtuins, were originally discovered in yeast for their role in prolonging replicative lifespan." Sir2 proteins were discovered for their role in silencing (hence the name Silent Information Regulator), and the roles in lifespan were only later reported.

2. Line 107-108 "both SIRT426 and SIRT627 are found to exhibit ADP-ribosyltransferase activity?" The authors should also mention the reported de-fattyacylation activity of SIRT6.

3. Line 111 "Therefore, there is a distinct possibility that the genuine enzymatic activity of SIRT proteins is still to be identified." This suggests that previously reported enzymatic activities of sirtuins are not genuine. The authors certainly do not intend to question the validity of an entire field, so they should remove this statement. It's likely they mean to say that additional novel activities may yet be identified, as in fact they do in this study.

4. The notion that SIRT7 is the "least studied" sirtuin is subjective (SIRT4 also very little studied) - and does not substantively add to understanding the biology or state of the field. They should leave out this type of phrasing.

Finally, the new text is in places very rough and needs considerable editing.a

Reviewer #3 (Remarks to the Author):

The authors have addressed my concerns. i would suggest to accept the paper as it is.

Response to reviewers' comments

Reviewer #1

In this revised manuscript, Yu and colleagues did major efforts to respond to the reviewers concerns, and the manuscript is highly improved. In particular the new data in other cells, the higher quality and the quantification of the westerns, and the clonogenic survival assays look very convincing. One minor comment, for fairness, it is surprising the authors do not include in their Discussion the fact that other sirtuins have been previously shown to be recruited to DBSs (see the papers from Tsai and colleagues on Sirt1 and Mostoslavsky and colleagues for SIRT6). Adding those to the Discussion, and putting Sirt7 in that context, will be reassuring. The manuscript warrant publication. and it will be of broad interest to researchers in the field of genomic instability and sirtuins.

Authors: The reports on the involvement of other sirtuins in DNA damage repair have been cited and discussed.

Reviewer #2:

Li et al, Nature Comm 2016

In this revised manuscript and response the authors have made good efforts to address the previous reviews, and they have adequately answered my concerns. The identification of SIRT7 as a histone desuccinylase is novel and of strong general interest. I have one final issue and some minor textual suggestions.

Major issue:

As indicated by another reviewer, the authors did not consider contributions of other SIRT7 substrates to the DNA damage response biology. In the revision, they have now added data showing possible additional desuccinylation sites targeted by SIRT7, and this is interesting. I also agree with the authors that given the lack of antibodies to these sites, further characterization of these sites is not necessary for this manuscript, which is focused on H3K122succ. However, SIRT7 does have additional substrates, namely H3K18Ac, several non-histone acetylated substrates, and

presumably additional ones yet-to-be identified. It is possible that some of the effects of SIRT7 on DNA damage responses and DNA repair might be mediated by such other substrates. While I believe analysis of such models is beyond the scope of this study, it is important that the authors include a brief discussion that other SIRT7 substrates might be involved in the DNA repair biology of SIRT7. This would not detract from the current findings, and in fact would add interest that SIRT7 may coordinate multiple aspects of DNA damage biology.

Authors: The issue has been discussed.

Minor issues:

The following introductory statements in the are inaccurate and should be revised:

1. Line 99: "Silent information regulator 2 (Sir2) proteins, or sirtuins, were originally discovered in yeast for their role in prolonging replicative lifespan." Sir2 proteins were discovered for their role in silencing (hence the name Silent Information Regulator), and the roles in lifespan were only later reported.

Authors: The statement has been modified accordingly.

2. Line 107-108 "both SIRT426 and SIRT627 are found to exhibit ADP-ribosyltransferase activity?" The authors should also mention the reported de-fattyacylation activity of SIRT6.

Authors: The report on de-fattyacylation activity of SIRT6 has been cited and discussed.

3. Line 111 "Therefore, there is a distinct possibility that the genuine enzymatic activity of SIRT proteins is still to be identified." This suggests that previously reported enzymatic activities of sirtuins are not genuine. The authors certainly do not intend to question the validity of an entire field, so they should remove this statement. It's likely they mean to say that additional novel activities may yet be identified, as in fact they do in this study.

Authors: The statement has been modified.

4. The notion that SIRT7 is the "least studied" sirtuin is subjective (SIRT4 also very little studied) - and does not substantively add to understanding the biology or state of the field. They should leave out this type of phrasing.

Authors: The sentence has been deleted.

Finally, the new text is in places very rough and needs considerable editing.

Authors: We have carefully edited the text.